



# Hydrographic section along 55°E in the Indian and Southern oceans

Katsuro Katsumata[1], Shigeru Aoki[2], Kay I. Ohshima[2], and Michiyo Yamamoto-Kawai[3]

[1]Graduate School of Science, University of Tokyo
[2]Institute of Low Temperature Science, Hokkaido University
[3]Department of Ocean Sciences, Tokyo University of Marine Science and Technology

**Correspondence:** Katsuro Katsumata (katsumata@eps.s.u-tokyo.ac.jp)

**Abstract.** A hydrographic section along 55°E, south of 30°S, was visited from December 2018 to January 2019 as the first occupation under the Global Ocean Ship-based Hydrographic Investigation Program. The water column was measured from the sea surface to 10 dbar above the bottom with eddy-resolving station spacings and the state-of-the-art accuracy. The upper profile was characterised by a conspicuous front between 42.5° and 43°S and a cold-core eddy at 39°S. The front was identified as

the confluence of Subtropical and Subantarctic fronts. The Agulhas Return Current front was found at 41.6°S. When combined with the section north of 30°S observed in 2018, another subsurface front was found in dissolved oxygen around 28°S at depths of 1500 to 3000 dbar. In the eastern Weddell-Enderby Abyssal Plain, no obvious mean flow was observed at depths greater than 3000 dbar. We used transient tracers to estimate isopycnal diffusivity there to be $72 \pm 16 \ \mathrm{m^2 s^{-1}}$. Antarctic Bottom Water in the basin consisted of water masses originating from the Cape Darnley region ($0-35\%$) and Weddell Sea Deep Water

($5-75\%$), diluted by Lower Circumpolar Deep Water above. These snapshot observations not only confirm hydrographic features reported earlier in the Madagascar and Crozet Basins, but also describe the diffusive nature of the deep to bottom circulation in the Weddell-Enderby Abyssal Plain. Some of the stations in the Crozet Basin were sampled in the 1980s and 1990s. Changes in the temperature-salinity relationship since then indicate warming of Upper Circumpolar Deep Water, volume reduction of Antarctic Bottom Water, and slight freshening which is stronger southward.

## 1 Introduction

Antarctic Bottom Water (AABW) produced around Antarctica spreads equatorward mainly along the western boundaries of the ocean basins. For this reason, hydrographic sections revealing the deep water masses along 50°E or 60°E in the Indian Ocean have often been discussed in the context of global-scale circulation (e.g. Johnson, 2008; Mantyla and Reid, 1983). Nevertheless, synoptic hydrographic observations of these sections (Warren, 1978; Park et al., 1993) are rare mainly because

of logistical difficulties. The GO-SHIP (Global Ocean Ship-based Hydrographic Investigations Program) section I07S is the latest occupation. It extends, for the first time, the existing section (I07N; equatorward of 30°S) to Cape Ann, the northernmost point in the East Antarctica (Fig. 1). The section follows the approximate flow path of AABW, and also crosses the Antarctic Circumpolar Current (ACC). When the ACC, which manifests as a bundle of fronts, flows between or over topographic features, mesoscale eddies are produced. The section crosses these fronts and eddies (Fig. 1). These fronts as well as water masses at

shallow and intermediate depths have been extensively studied in a series of cruises by Y.-H.Park (Park et al., 1991, 1993;



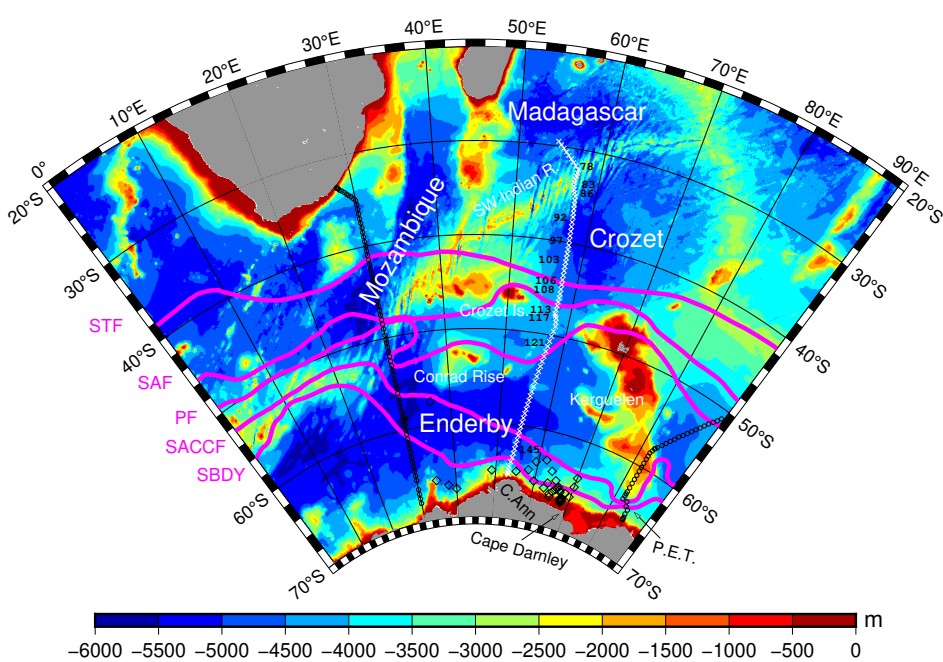

**Figure 1.** ETOPO5 bathymetry (colour scale) with basin names and topographic features, C.Ann (Cape Ann), and P.E.T. (Princess Elizabeth Trough). The white crosses between 50°E and 60°E indicate the I07S section; the black circles at 30°E indicate the I06S section; and those between 80°E and 90°E indicate the I08S section (stations occupied in 2007). Selected stations along the I07S section are labelled. The southernmost station is 153. The diamonds near the coast of Antarctica are R/V *Hakuho* (KH) cruise stations. The magenta lines show the climatological positions of the five major fronts as defined by Orsi et al. (1995); Subtropical Front (STF), Subantarctic Front (SAF), Polar Front (PF), Southern Antarctic Circumpolar Current Front (SACCF), and Southern Boundary (SBDY).



Park and Gambéroni, 1997; Park et al., 1998) and by later cruises (e.g. Meijers et al., 2010; Williams et al., 2010; Jullion et al., 2014; Ryan et al., 2016). In contrast, the deep circulation within the Weddell-Enderby Abyssal Plain has received limited attention. In this paper, for convenience, we call the eastern part of the Abyssal Plain as the Enderby Basin. The northern limb of the gyral circulation within the Enderby Basin is an eastward flow along the southern flank of the Southwest Indian Ridge;

Gordon and Huber (1984) called this flow as the Weddell Cold Regime because the water mass is mixed with cold winter water produced along the coastal shelves of the Weddell Sea. This cold water mixes with relatively warm water carried by the ACC to form the Weddell Warm Regime, which flows westward as the southern limb of the Weddell Gyre. The eastern boundary of the Weddell Gyre has not been clearly defined by observation data. Direct measurement of velocity by a lowered Acoustic Doppler Current Profiler (Meijers et al., 2010) suggests that the boundary is close to 40°E. Jullion et al. (2014) used a box

inverse model to estimate transports across the I06S section (30°E). They found the zonal transport in deep to bottom layers to be mostly eastward along the southern flank of the Southwestern Indian Ridge, but it includes both eastward and westward components in the interior of the Enderby Basin. Further south near the coast, the transport is dominated by a westward flow along the Antarctic Slope Front. It is not clear how these structures extend eastward to the I07S section at 55°E. Here, we examine the deep circulation in the Enderby Basin as revealed by hydrographic data collected along the section where it passes

through the basin.

As the subtropical frontal jet of the ACC negotiates the Southwest Indian Ridge around (45°E, 40°S), vigorous eddies are produced to leeward over the Crozet Basin (Fig. 2). At 1000 dbar depth, eddy activity is also seen at around (30°E, 50°S) and its effects extend to the Enderby Basin to leeward. The difference between the surface and 1000 dbar might be attributable to the choice of contour values, and to weaker stratification over the Enderby Basin, or both. Our analysis shows that, because of

these eddies, the deep circulation in the Enderby Basin is diffusive (see Section 4.1).

Further south along the Antarctic coast, the I07S section crosses the westward flowing Antarctic Slope Front (not shown in Fig. 1). Considering data from a transient tracer section across the Weddell Gyre, Meredith et al. (2000) predicted the existence of a deep water source at around 75°E, upstream of the Antarctic Slope Front. Later, Ohshima et al. (2013) identified a deep water source near Cape Darnley (Fig. 1). Turning to numerical simulations, we find that a tracer ("brine") release experiment by

Kusahara et al. (2017) has shown that the bottom water in the Enderby Basin is mostly from the Cape Darnley Polynya. Initially, the tracer spread westward over the continental shelf; it then circulates the Weddell Gyre before continuing equatorward into the Crozet Basin. There also appeared to be a direct path from Cape Darnley to the Crozet Basin (Figure 10d of Kusahara et al., 2017). The direct spread of fresh Cape Darnley Bottom Water (CDBW) to the Enderby Basin is consistent with the result of mixing budget analyses using near-bottom hydrographic data by Aoki et al. (2020a) and Gao et al. (2022). This behaviour of

the bottom water is also consistent with the observed property distribution (Figure 8b of Orsi et al., 1999); the temperature-salinity characteristics at bottom water densities was much fresher within the longitude band of 30°E to 60°E than those within 60°E to 90°E. In Section 4.2, we estimate the mixing ratio of Cape Darnely Bottom Water and Weddell Sea Deep Water.

When planning the cruise along I07S, whenever possible we attempted to place the stations so that they overlapped existing stations for which high quality data were available. Decadal variability in water properties at these stations is discussed in

Section 5.

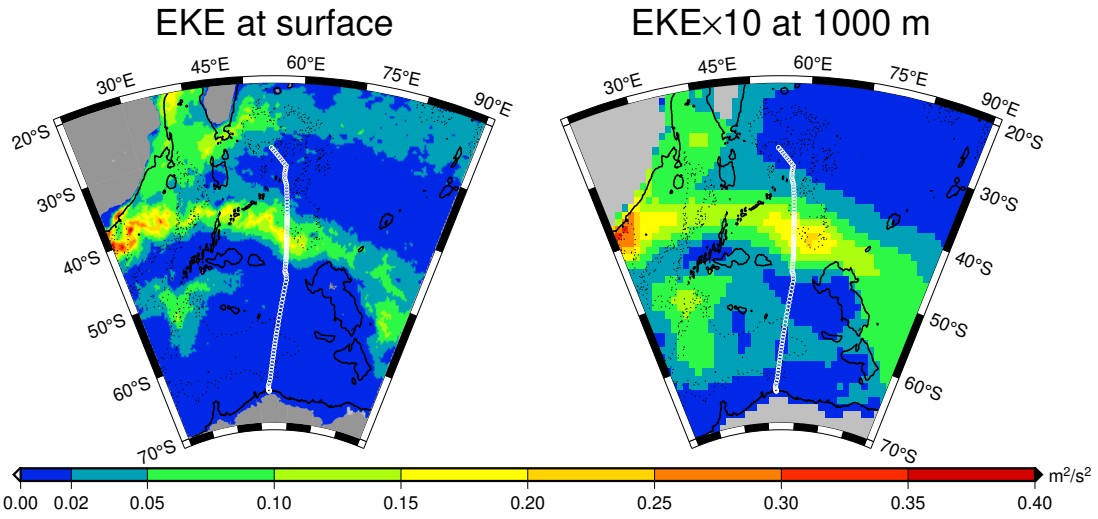

**Figure 2.** Eddy kinetic energy (EKE) at the surface (left) and at 1000 dbar depth (right). The surface EKE was estimated from geostrophic velocity anomalies of the altimetry data on a 0.25° grid. Values at 1000 dbar were estimated from the Scripps Argo float drift data on a 1° grid. The mean field was defined as the spatial and temporal average within a circle of radius 300 km with the centre at the grid point. See Katsumata (2017) for details of the EKE calculation at 1000 dbar. Solid black and dotted bathymetry contours indicate 5000 m and 2000 m depth, respectively.





## 2 Data

The primary data used here are from conductivity, temperature, and depth profiler (CTD) and bottle measurements made onboard R/V *Mirai* during the cruise from 31 December 2019 (first station at 29.5°S) to 22 January 2020 (last station near seasonal ice edge at 65.3°S), nominally along 55°E (Uchida et al., 2021). By using the reference thermometer (SBE35) and bottle data, the CTD data comply the GO-SHIP standard accuracies of 0.002°C (temperature), 0.002 g/kg (salinity), and 1% (dissolved oxygen) (Hood, 2010). A pair of lowered Acoustic Doppler Current Profilers (LADCP) – one looking upward and the other downward – was used simultaneously. We also use data from two other GO-SHIP sections across the ACC; one section is located to the south of Africa along 30°E (section I06S), and the other crosses the Princess Elizabeth Trough nominally along 84°E (section I08S). Data collected at several stations during cruises of M/V *Marion Dufresne* (MD43, MD68, and MD75) and GO-SHIP I05 were used for decadal comparisons (see Section 5). CTD and bottle data from the latest cruises of R/V *Hakuho* near Cape Darnley (KH19 and KH20) were also used. Additionally, gridded sea surface altimeter data from the Copernicus Marine Service and Argo float drift data from the Scripps Institution of Oceanography were used (see Data Availability section). To denote density, we use $\gamma^n$ notation where $\gamma^n = 26.8$ means a neutral density (Jackett and McDougall, 1997) of 1026.8 $\mathrm{kgm^{-3}}$.

## 3 Fronts

Park et al. (1991, 1993) defined four fronts that they observed in the Crozet Basin by their hydrographic characteristics at 200 dbar depth (Figure 4). These front definitions as they apply to the I07S section are summarised in Table 2, which also lists the major southerly fronts defined by Orsi et al. (1995). The Southern ACC Front is found at around 62.5°E. At around 64.8°E, the Southern Boundary of the ACC passes over the continental slope at about 3000 dbar depth. These locations are consistent with those reported in Williams et al. (2010) who indicated that the Weddell Gyre is limited to the area west of Cape Ann. Hydrographic sections of temperature, salinity, and dissolved oxygen are shown in Fig. 3. The station interval of 30 nautical miles (15 nm with XCTD) was found to be insufficient to separate the Subtropical and Subantarctic fronts. The low salinity and low temperature anomaly centred at 39°S (about $X =2200$ km) is a cyclonic (negative sea surface height anomaly) eddy, not a front. As seen in Fig. 5, our observation track passed very close to the eddy's centre, and the upward heave of isotherms and isohalines are striking. The real fronts are found located around 41°S to 43°S, ($X =2400$ to 2600 km) (Fig. 5). The three Northern fronts are accompanied by a strong eastward jet with a vertically averaged velocity $> 0.1$ $\mathrm{ms^{-1}}$ but the jet along the Polar Front is weaker at $< 0.03$ $\mathrm{ms^{-1}}$ Comparison of two snapshots obtained 4 days apart (Fig. 5) demonstrates that some eddies near the front (e.g., around 57.5°E, 43.5°S) were advected eastward, whereas others (e.g., 58°E, 39°S) moved westward, in agreement with the description by Park and Gambéroni (1997) that "even a section made in a few days across the frontal zone might not be considered as synoptic".

The slopes of the isohaline and isotherms at the fronts are steep: $S_A = 34.82$ g/kg isohaline is at 560 dbar at $X = 2591.2$ km and it crops out north of the next station at $X = 2616.4$ km, whereas the $\Theta = 9$°C isotherm jumps from 555 dbar to 45



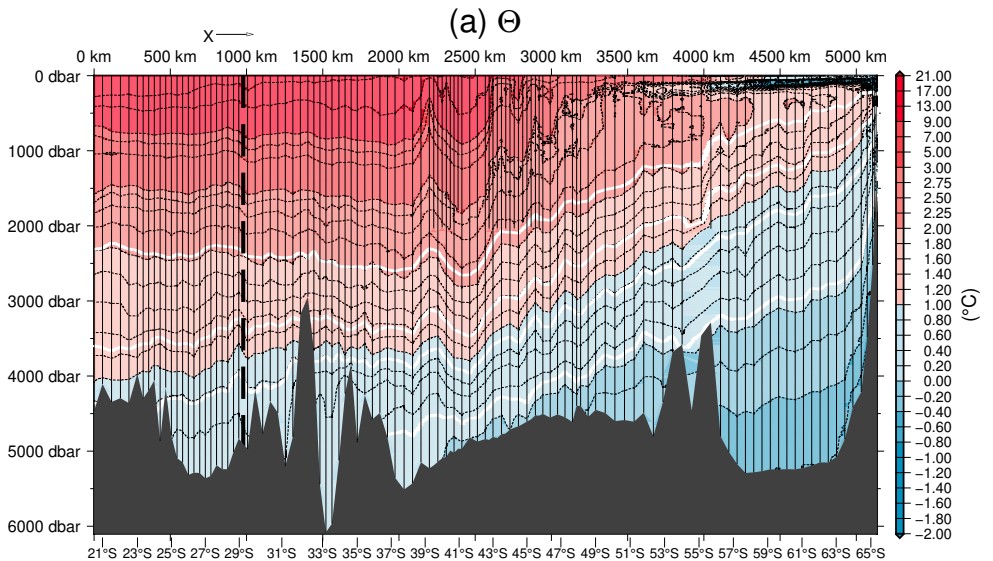

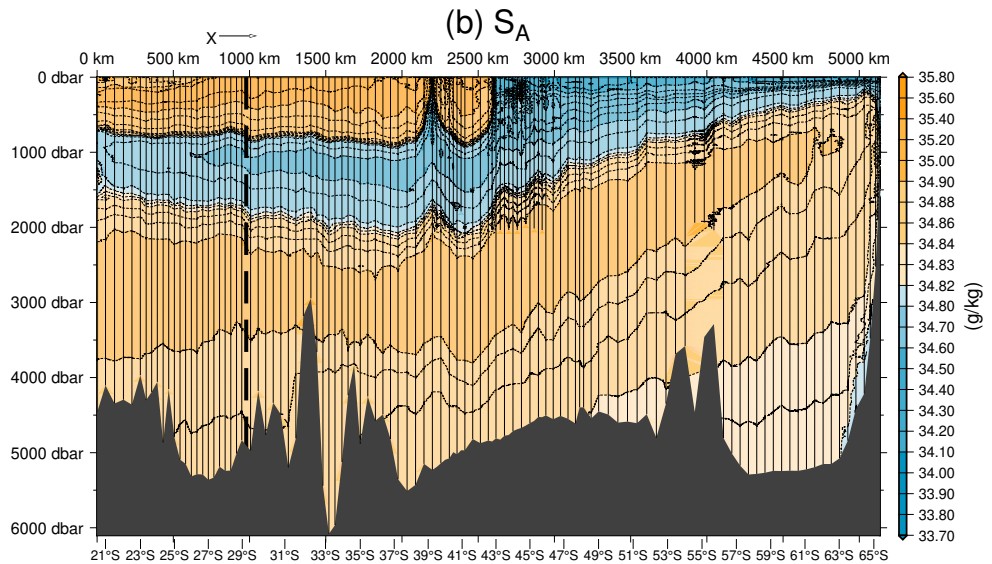

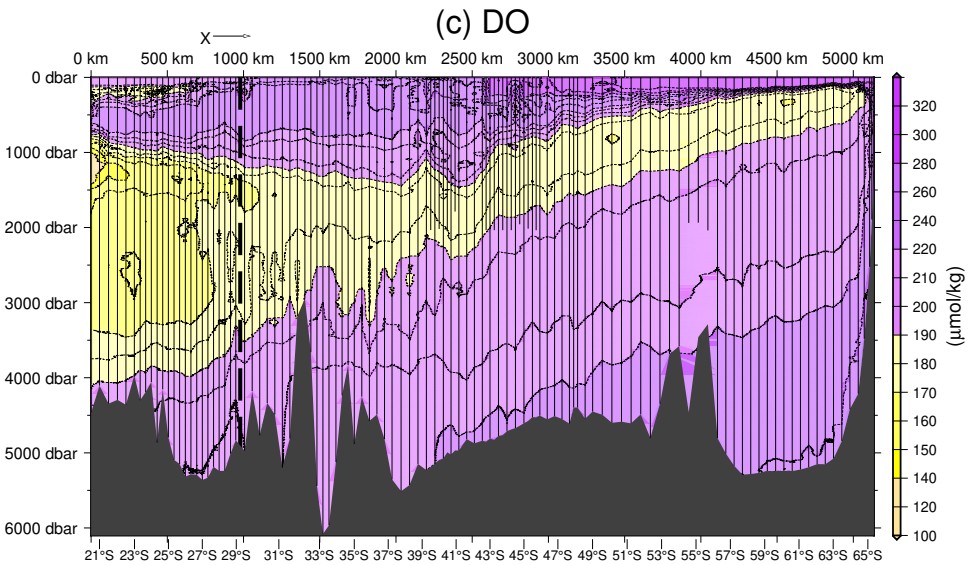

**Figure 3.** (a) Conservative Temperature $\Theta$ (°C), (b) Absolute Salinity $S_A$ (g kg$^{-1}$), and (c) Dissolved Oxygen ($\mu$mol kg$^{-1}$) observed along the I07 section. The section north of 29.5°S (vertical dashed line) was occupied from April to May 2018 during cruise 33RO20180423, and that south of 29.5°S was occupied from December 2019 to January 2020 during cruise 49NZ20191229. The thin black lines show the CTD traces. Those found only above 2000 dbar are expendable CTD (XCTD) traces. The upper horizontal axis shows distance ($X$) from Station 26 of cruise 33RO20180423 at 54.517°E, 20.502°S. The vertical axis shows depth. The bottom axis shows approximate latitude. In (a), the white contours indicate neutral densities of $\gamma^n = 28.0, 28.11, 28.18$, and $28.27$. Triangular shapes found at the ends of XCTD traces (e.g. around $X = 4000$ km at 2000 dbar in (a) and (b) and 4000 dbar in (c)) are interpolation noise induced during plotting.





**Table 1.** Cruises. The cruises for sections I07S, I07N, I06S, I08S, and I05 are under the Global Ship-based Hydrographic Investigation Program (or its predecessor World Ocean Circulation Experiment). KH19 and KH20 are cruises onboard R/V *Hakuho* (Ohashi et al., 2022). Others are cruises onboard M/V *Marion Dufresne* (e.g. Park et al., 1991).

| Section | Cruise duration | `Expo code` or PROJECT |
|---------|-----------------|------------------------|
| I07S | Dec 2019 – Feb 2020 | `49NZ20191229` |
| I07N | Apr 2018 – Jun 2018 | `33RO20180423` |
| I06S | Apr 2019 – May 2019 | `325020190403` |
| | Feb 2008 – Mar 2008 | `33RR20080204` |
| | Feb 1996 – Mar 1996 | `35MF103_1` |
| | Feb 1993 – Mar 1993 | `35MFCIVA_1` |
| I08S | Feb 2016 – Mar 2016 | `33RR20160208` |
| | Feb 2007 – Mar 2007 | `33RR20070204` |
| | Dec 1994 – Jan 1995 | `316N145_5` |
| I05 | Mar 2009 – May 2009 | `33RR20090320` |
| | Mar 2002 – Apr 2002 | `74AB20020301` |
| | Nov 1987 – Dec 1987 | `74AB29_1` |
| KH19 | Jan 2019 – Feb 2019 | – |
| KH20 | Jan 2020 – Feb 2020 | – |
| MD43 | Mar 1985 | INDIGO |
| MD68 | Apr 1991 | SUZIL |
| MD75 | Apr 1993 | ANTARES |

**Table 2.** Major fronts at $55°$E. The traditional potential temperature and practical salinity scale are used in this table to facilitate comparison with previous publications.

| Front | $\theta_0$ range | axial value | $S_p$ range | axial value | station | latitude |
|-------|-----------|-------------|-------------|-------------|---------|----------|
| Agulhas Return Current[a] | 12–16 | 14 | 35.1–35.5 | 35.3 | 100 | 41.6°S |
| Subtropical[a] | 8–12 | 10 | 34.6–35.1 | 34.8 | 102 or 103 | 42.5 or 43°S |
| Subantarctic[a] | 4–8 | 6 | 34.1–34.5 | 34.3 | 103 | 43°S |
| Polar[b] | – | – | – | – | 121 | 51°S |
| Southern ACC[c] | – | – | – | – | 145 | 62.5°S |
| Southern Boundary[d] | – | – | – | – | 150 | 64.8°S |

a As defined by Park et al. (1991) using temperature and salinity at 200 dbar depth.

b Northern limit of subsurface potential temperature minimum of $\theta_0 = 2°C$.

c Potential temperature maximum $\theta_0 > 1.8°C$ (Orsi et al., 1995).

d Potential temperature maximum $\theta_0 > 1.5°C$ (Orsi et al., 1995).





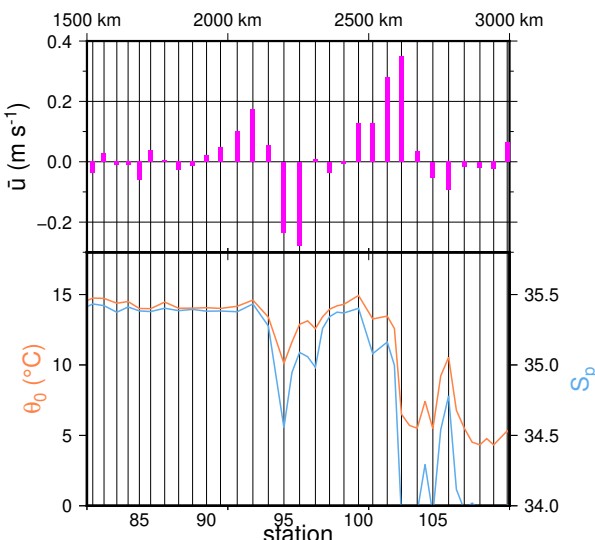

**Figure 4.** Zonal velocity averaged over the water column (upper panel), and potential temperature ($\theta_0$, orange) and practical salinity ($S_p$, cyan) at 200 dbar depth. The zonal velocity is positive eastward. The traditional potential temperature and practical salinity scale are used in this figure to facilitate comparison with previous publications.

dbar between these stations. Both have a slope of $2 \times 10^{-2}$. Because the meridional gradients of temperature and salinity are density compensating, the slope of the corresponding isopycnal is smaller (506 to 271 dbar or $0.2 \times 10^{-2}$).

The steep salinity front can be explained by surface forcing. The meridional gradient of surface salinity in the climatological data set (Reagan et al., 2024) across these fronts (Fig. 6) is the strongest in the Southern Hemisphere. Strong evaporation in the midlatitude Indian Ocean results in salty surface mixed layer. The sality surface water mass is advected by the Agulhas Current and returns in the Agulhas Return Current to the north of the fresh ACC, which amplifies the meridional gradient of precipitation minus evaporation.

Using data from a limited number of stations, Park et al. (1993) mapped dissolved oxygen at 3000 dbar depth (their Figure 11) and found a frontal structure between (57°E, 32°S) and (75°E, 43°S), separating low-oxygen North Indian Deep Water (NIDW) to the northeast from oxygen-rich Lower Circumpolar Deep Water (LCDW) to the southwest. Across the front, strong



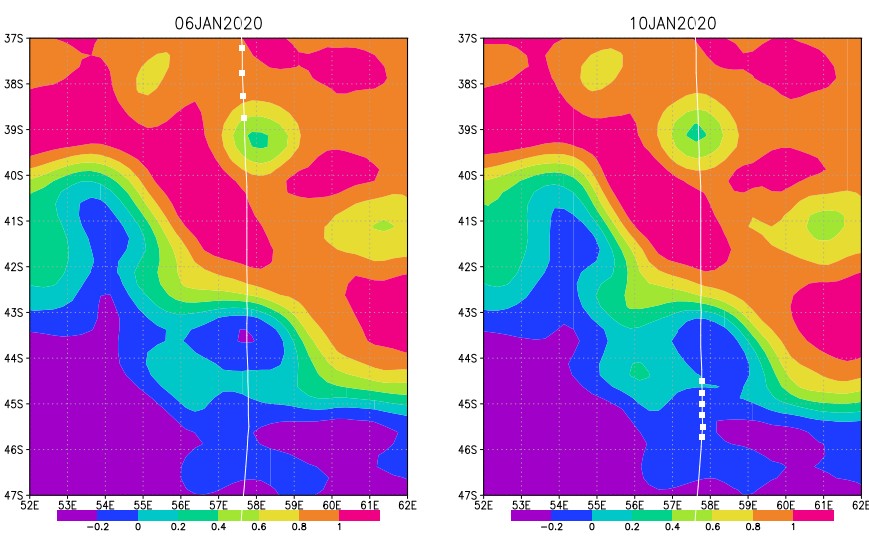

**Figure 5.** Two snapshots of the absolute dynamic topography; 6 January 2020 UTC (left) and 10 January 2020 UTC (right). The thin white line shows the cruise track, and the white squares show the CTD and XCTD stations occupied on each day.

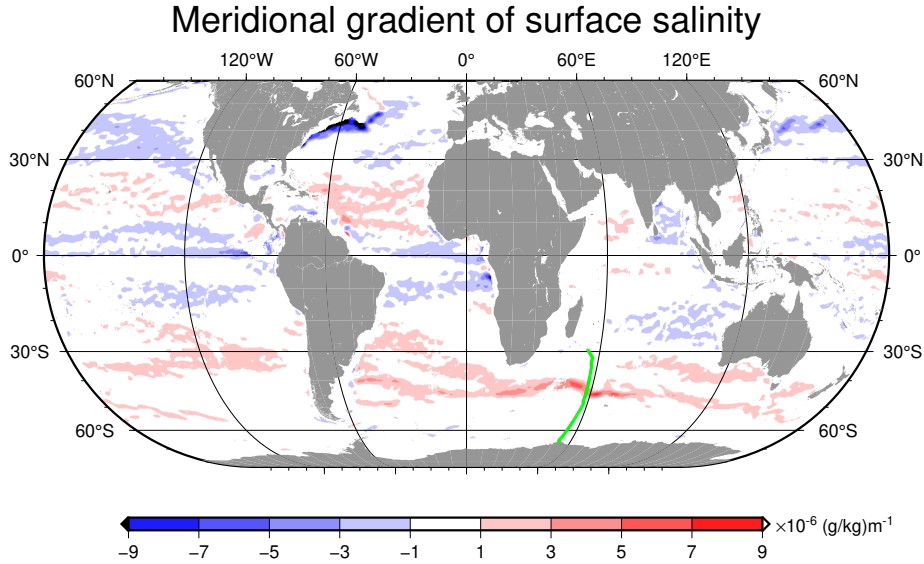

**Figure 6.** Meridional gradient of surface salinity from the World Ocean Atlas 2023 (Reagan et al., 2024). The meridional gradient was calculated from the gridded salinity product on a $\frac{1}{4}$-degree grid. Large cross-shore gradients in coastal regions are avoided by restricting the plot to ocean depths > 3000 m. The Arctic Ocean is also excluded. Green dots indicate the I07S stations.

isopycnal mixing of NIDW and LCDW was inferred from the water properties. Our section (Fig. 3c) which crosses the front at around 28°S (which was observed as part of I07N) and shows the corresponding mesoscale structures north of 33°S at around 3000 dbar depth, is supportive of vigorous isopycnal mixing.

## 4    Deep circulation in the Enderby Basin

A snapshot of the horizontal circulation in the deep Enderby Basin is examined by plotting the LADCP data (Figure 7). As suggested by the mean velocities, which are well within the variance ellipses, there is no obvious mean flow. As the mesoscale wiggles of the sea surface height contours suggest, the field is characterised by rich eddies, and our snapshot observation could not capture any mean transport. The eddies imply that the circulation in the deep Enderby Basin is diffusive in nature.







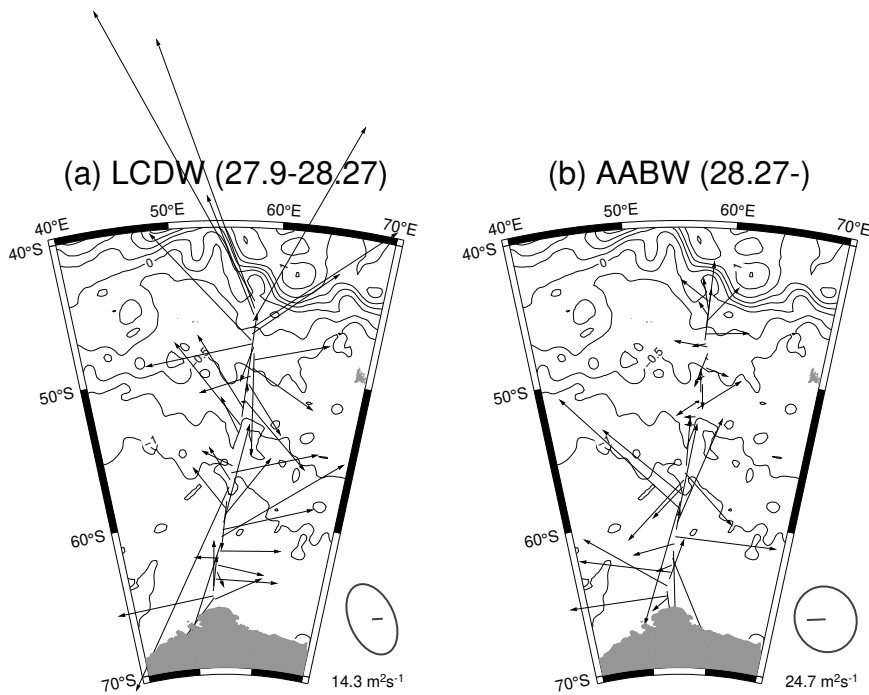

**Figure 7.** Instantaneous velocity observed by LADCP. (a) Horizontal velocity averaged within the LCDW layer ($27.9 < \gamma^n < 28.27$) and (b) within the AABW layer ($28.27 < \gamma^n$). The contours with intervals of 0.25 m (thin) and 0.5 m (thick, labelled) show the absolute sea surface height (SSH) field averaged over the cruise period (31 December 2019 to 22 January 2020). The SSH data are from satellite altimetry. The variance (ellipses) and mean (vectors) of the horizontal velocity in the Enderby Basin (defined here as south of $56°$S) are shown at the right bottom corner of each panel. The vectors also give the scale.

It is possible to estimate vertical diffusivity from LADCP and CTD data by using a finescale internal wave parameterisation as is discussed in detail by Sasaki et al. (2024). We found that vertical diffusivity in the deep Enderby Basin is moderately strong ($10^{-5}$ to $10^{-4}$ $\text{m}^2\text{s}^{-1}$) and is enhanced towards the bottom. Using this vertical diffusivity, we attempted to estimate the horizontal diffusivity from the spatial distribution of transient tracers.

**4.1 Isopycnal diffusivity estimated from transient tracer distributions**

Four kinds of transient tracers were measured along I07S. Here, we used CFC12 and SF$_6$ because CFC12 has had the highest atmospheric concentration since the 1970s and because time evolution of SF$_6$ is quite different from that of CFC12.



As mentioned in Section 4, no obvious mean flow was observed. Without any advecting mean flow, the concentration of a tracer in a meridional section $c(y,z,t)$ is governed by

$$\frac{\partial c}{\partial t} = K\frac{\partial^2 c}{\partial y^2} + D\frac{\partial^2 c}{\partial z^2}, \tag{1}$$

where $t$ is time; the direction of $y$ and $z$ are northward along the isopycnal surfaces and across the isopycnal surfaces (positive towards the lighter density), respectively (McDougall, 1984); $K$ is isopycnal diffusivity; and $D$ is diapycnal diffusivity. Spatial and temporal variations of $K$ and $D$ are neglected. If we scale time by $T$, horizontal distance by $\sqrt{KT}$, and vertical distance by $\sqrt{DT}$, the equation becomes simply

$$\frac{\partial c}{\partial t} = \frac{\partial^2 c}{\partial y^2} + \frac{\partial^2 c}{\partial z^2}. \tag{2}$$

Profiles of the tracers (Figure 8b and 8e) clearly show that dense shelf water with high tracer concentrations flows along the continental slope of Antarctica. We approximate this slope as a vertical wall and model this situation by considering two time-dependent tracer sources: one is a point source at the bottom of the slope and the other is a line source placed along the southern vertical wall. The point source represents young dense water masses carried westward by the Antarctic Slope Current, possibly including some contribution from the Australia Antarctic Basin through the Princess Elizabeth Trough, whereas the line source is the newly ventilated water flowing along the continental slope. In neutral density coordinates (Figure 8a and 8c), the point source is at $(y,z) = (0,0)$ and the line source is at $y = 0$. We designate tracer distribution originating from the point source as $c_0$ and that originating from the line source as $c_1$ and seek a solution as their superposition;

$$c(y,z,t) = \alpha_0 c_0(r,t) + \alpha_1 c_1(y,t), \tag{3}$$

where $\alpha_0$ and $\alpha_1$ are constants to be determined. The solution $c_0(r,t)$ is governed by (2) and is given in cylindrical coordinates with $r = \sqrt{y^2 + z^2}$:

$$\frac{\partial c_0}{\partial t} = \frac{1}{r}\frac{\partial c_0}{\partial r} + \frac{\partial^2 c_0}{\partial r^2}, \tag{4}$$

with the boundary conditions

$$c_0(r=0,t) = s_0(t-\delta_0) \tag{5}$$

$$\frac{\partial c_0(y=0,t)}{\partial z} = \frac{\partial c_0(z=0,t)}{\partial y} = 0, \tag{}$$

where the time offset $\delta_0$ is the advection time from the surface mixed layer to the I07S section, and $s_0$ is the source of the tracer as a function of time. The solution $c_1(y,t)$ is governed by

$$\frac{\partial c_1}{\partial t} = \frac{\partial^2 c_1}{\partial y^2}, \tag{6}$$

with the boundary conditions

$$c_1(y=0,t) = s_1(t-\delta_1) \tag{7}$$

$$\frac{\partial c_1(y=0,t)}{\partial z} = 0, \tag{}$$



where $\delta_1$ is, again, the advection time, which can be different from $\delta_0$, and $s_1$ is the source function. The sum $\alpha_0 c_0 + \alpha_1 c_1$ only approximately satisfies the boundary condition at the southern boundary $y = 0$, which is a limitation of this model. The governing Eqs. (4) and (6) were numerically solved with realistic source functions $s_0(t)$ and $s_1(t)$. We take the unit time as $T = 1$ year. Calculations were carried out for the year 1950 to 2020 with a non-dimensional grid spacing of $0.1$ for $0 < r < 50$ and $0 < y < 50$. The source functions reflect the atmospheric concentration of the tracers, which is converted to the concentration within the mixed layer by laboratory-determined solubility (Warner and Weiss, 1985; Bullister et al., 2002) at an assumed seawater temperature of $0°$C, salinity of $34.5$ practical salinity unit, and a constant saturation rate. A typical observed saturation rate is 70% (Schlosser et al., 1991; Ohashi et al., 2022), but the tracers in the dense shelf water are diluted when they form the source water at depth. We tested saturation rates between 35% and 70% and found an overall error similar to that for saturation rates between 50% and 70%, but greater for smaller saturation rates. We therefore use a saturation rate of 50% here. The atmospheric concentration before 2015 is taken from Bullister and Warner (2017) and that after 2015 from the NOAA Global Monitoring Laboratory, following Ohashi et al. (2022).

Nonlinear least squares fit of Eq.(3) (Levenberg-Marquardt method, Madsen et al. (2004)) was used to find the values of $\alpha_0$, $\alpha_1$, $\delta_0$, $\delta_1$, and $K$ that minimise the difference between the predicted and observed concentration of CFC12 and SF$_6$. In the calculation of this difference, the source area was excluded by by limiting the comparison to observed concentrations of $< 0.4 \, \text{pmolkg}^{-1}$ for CFC12 and $< 0.21 \, \text{fmolkg}^{-1}$ for SF$_6$; thus we fit Eq.(3) to 125 and 118 bottle data for CFC12 and SF$_6$, respectively. To estimate the isopycnal diffusivity $K$, we start from the fixed diapycnal diffusivity of $D = 4.7 \times 10^{-5} \, \text{m}^2\text{s}^{-1}$ which is the arithmetic mean of the diapycnal diffusivity below 2000 dbar between $56°$S and $63.5°$S estimated by finescale parameterisation (Polzin et al., 2014) of the observed stratification and horizontal velocity field (Katsumata et al., 2021). The observed diapycnal diffusivity showed some spatial variability (Fig. 8c) and the diffusivity is high where the near-bottom CFC12 concentration is high ($Y = 400$ to $700 \, \text{km}$ in Fig. 8a and b). We neglect this spatial variability. Next, we prescribe a set of initial values, namely, at $\alpha_0 = 0.1, \alpha_1 = 0.1$, and $K = 500 \, \text{m}^2\text{s}^{-1}$ for fitting by the Levenberg-Marquardt method. The advection times $\delta_1$, $\delta_0$ are each set at $0, 1, \cdots 19$ years, giving 400 combinations of $\delta_1$ and $\delta_0$ for the nonlinear least squares calculation; thus we obtain 400 sets of $\alpha_1$, $\alpha_2$, and $K$. We tried several different sets of initial values (for example $K = 100$ $\text{m}^2\text{s}^{-1}$) but they converged to the same solution.

The error did not show an obvious sharp global minimum. The global minimum error (sum of squares) was 0.117 for the 400 set of trials and the error histogram (not shown) is a decreasing function of the error with a change of the decreasing slope at error $= 0.25$. We thus selected the 160 parameter sets with error $< 0.25$ as good estimates. The means and standard deviations of the good estimates are are $\delta_0 = 10 \pm 6$ years, $\delta_1 = 16 \pm 3$ years, $\alpha_0 = 0.12 \pm 0.04$, $\alpha_1 = 0.14 \pm 0.01$, and $K = 72 \pm 16 \, \text{m}^2\text{s}^{-1}$. These mean values are used in Figure 8. The "age" of the source water ($\delta_0$ and $\delta_1$) appears to be overestimated compared with ages obtained by other methods e.g. Ohashi et al. (2022); possible reasons include the saturation uncertainty or a limitation due to the idealised source function.

The estimated isopycnal diffusivity of $K = 72 \pm 16 \, \text{m}^2\text{s}^{-1}$ is not unlike the deep-sea value. For example, the isopycnal diffusivity at 4000 dbar in the Atlantic Ocean has been estimated by an artificial tracer release experiment to be about 100 $\text{m}^2\text{s}^{-1}$(Ledwell, 2024). Phillips and Rintoul (2000) have estimated horizontal temperature diffusivity near the Subantactic

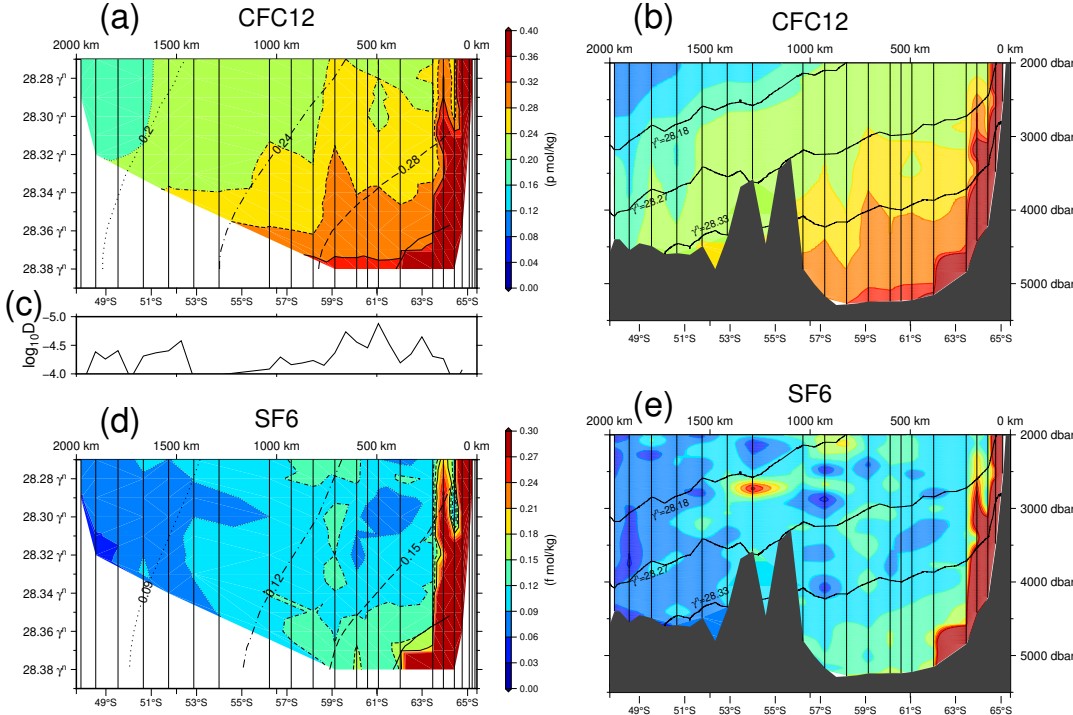

**Figure 8.** Concentrations of transient tracers (a, b) CFC12 and (d, e) $SF_6$. The horizontal axes show latitude (bottom) and distance ($Y$, top) from the southernmost station, and the vertical axis shows density in (a) and (d) and depth in (b) and (e). The colours in (a) and (b) show the concentration of CFC12 in $pmol kg^{-1}$, whereas those in (d) and (e) show the concentration of $SF_6$ in $fmol kg^{-1}$. The contours in (a) and (d) show the concentrations predicted by the simple diffusion model (3) with $\alpha_0 = 0.12$, $\delta_0 = 10$, $\alpha_1 = 0.14$, and $\delta_1 = 16$. The same type of contour lines are used for observed and predicted quantities. The contours in (b) and (e) show the isopycnals $\gamma^n = 28.18, 28.27$, and $28.33$. The observed concentrations in the pressure vertical coordinate in (b) and (e) are converted to density vertical coordinate in (a) and (d). (c) shows diapycnal diffusivity $D$ averaged below $2000$ dbar.



front, South of Australia, to be 100 to 200 $\mathrm{m^2s^{-1}}$ at 2240 dbar and 3320 dbar. Our value is also similar to the isopycnal diffusivity on the continental rise in the Antarctic Australia Basin (Katsumata and Yamazaki, 2023) and a deep diffusivity of 30 to 70 $\mathrm{m^2s^{-1}}$ has been estimated by a plume model of AABW based on observation of transient tracers in the Weddell Sea (Haine et al., 1998). The diffusive nature of the interior circulation also agrees with the CFC distribution in Fine et al. (Figure 5 of 2008) with the contours much farther apart in the interior of the Enderby Basin than in near-boundary regions. The tracer distribution thus suggests that circulation in the deep Enderby Basin is mostly dominated by the westward transport near the Antarctic slope and isopycnal interior diffusion. The diffusivity value is typical of that for deep basins, where values are weaker than those observed in the upper water column (500 to 1000 $\mathrm{m^2s^{-1}}$), e.g. (Tulloch et al., 2014).

## 4.2 AABW composition

The Enderby Basin collects AABW from four major sources – Weddell Sea Deep Water (WSDW), Cape Darnley Bottom Water (CDBW), Adélie Land Bottom Water (ALBW), and Ross Sea Bottom Water (RSBW). WSDW arrives at the I07S section from the west, while ALBW and RSBW flow through the Princess Elizabeth Trough further to the east (Heywood et al., 1999).

Previous occupations of GO-SHIP Sections I06S at 30°E and I08S at 85°E provide information on how WSDW and ALBW+RSBW each contribute to AABW at I07S (Figure 9). On the temperature-salinity diagram, points for upper AABW ($28.18 < \gamma^n < 28.33$) on I08S are scattered (Fig. 9) because these densities on the shelf are subject to direct atmospheric forcing. The deep water masses are, however, similar to I07S within the AABW range ($28.27 < \gamma^n < 28.34$). Menezes et al. (2017) have reported freshening of bottom water in the Princess Elizabeth Trough. On I06S, points for the lighter water masses $\gamma^n < 28.22$ are similarly scattered. On I06S below $\gamma^n = 28.22$, persistent freshening is discernible (Couldrey et al., 2013) and the water on I07S appears to be in between the 2008 and 2019 water masses observed along I06S. Haine et al. (1998), using CFCs, estimated the mean transient time from the Greenwich meridian (0°E) to the Crozet-Kerguelen gap (55°E) along the southern flank of the Southwest Indian Ridge as 12 to 20 years (their Figure 5) depending on the diffusion assumptions adopted. The transient time between 30°E and 55°E is likely half that, i.e., 6 to 10 years. We therefore use the 2008 observations on I06S as the source water in our mixing model.

We note that the bottom water on I08S was not dense enough to explain the near-bottom water ($> 28.34$) observed at I07S. We thus conclude that the densest class of AABW ($\gamma > 28.34$) observed on I07S is a mixture of WSDW and CDBW.

Orsi et al. (1999) called the water mass with $28.18 < \gamma^n < 28.27$ "ACC Bottom Water" (ACCbw) because it is well mixed and showing little zonal variability compared to the water masses above or below. The focus in this section is thus on water masses deeper than $\gamma^n = 28.27$, which fill the Enderby Basin and south of 39°S, the Crozet Basin (Fig. 3a). Here we estimate the composition of AABW ($\gamma^n > 28.27$) in the Enderby Basin using a method following that used in Johnson (2008), i.e. least squares fitting with a non-negative constraint in which temperature, salinity, $\mathrm{PO^*} = 170[\mathrm{PO_4}] + [\mathrm{O_2}]$, $\mathrm{NO^*} = 10.625[\mathrm{NO_3}] + [\mathrm{O_2}]$, and $\mathrm{SO^*} = [\mathrm{SO_4}] + [\mathrm{O_2}]$ are conserved. The method yields six equations (one being mass conservation). We do not expect potential vorticity be conserved because the flow is subject to bottom friction. The weights for the conservation equations are taken from Johnson (2008); 1, 0.25, 1, 0.5, 0.5, 0.25 for mass, temperature, salinity, $\mathrm{PO^*}$, $\mathrm{NO^*}$, and $\mathrm{SO^*}$, respectively.



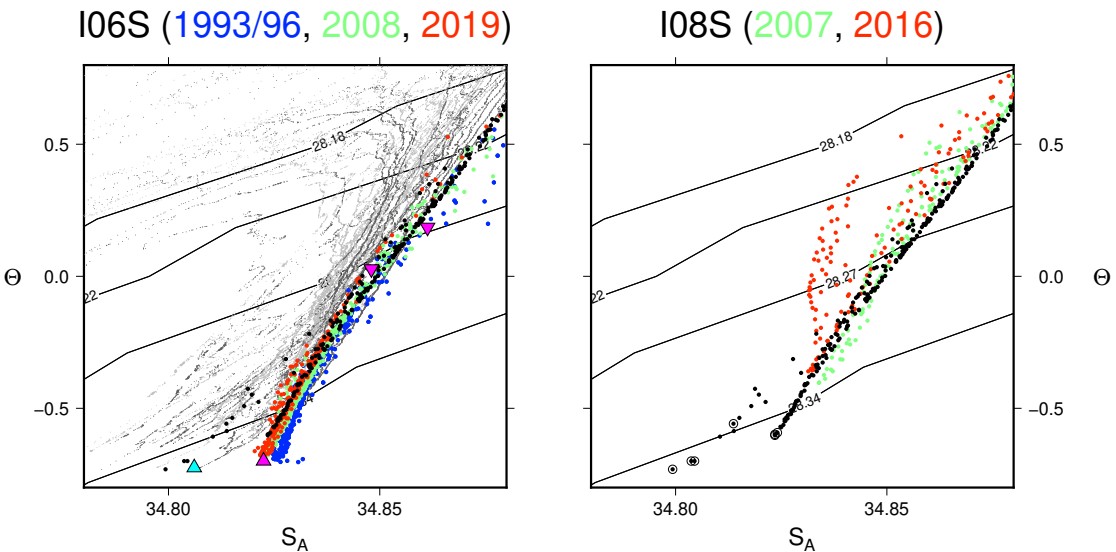

**Figure 9.** Temperature-salinity diagrams of the bottom waters at 30°E (I06S, left) and at 82°E (I08S, right) based on data from different occupation years (indicated by the colours of the circles). Black points show the data of I07S observed in 2019/20, and the dark and light grey points show data of R/V *Hakuho* (KH) cruises in 2019 and 2020, respectively (see Fig. 1). For I06S, only bottle data collected at depths with density $\gamma^n > 28.0$ are shown. For I08S, only bottle data from south of 60°S at depths with density $\gamma^n > 28.0$ and deeper than 1000 dbar are shown. For I07S, only bottle data at depths with density $\gamma^n > 28.0$ and deeper than 3000 dbar are shown. In the right panel showing I08S, data points for the deepest bottles (i.e. about 10 m above the seafloor) and south of 60°S along the I07S section (black points) are circled. The source waters used in the mixing analysis are indicated by triangles; inverted magenta triangles, LCDWs ($S_A = 34.86$ g/kg, $\Theta = 0.19°$C; $S_A = 34.85$g/kg, $\Theta = 0.03°$C); magenta triangle, WSDW ($S_A = 34.82$ g/kg, $\Theta = -0.70°$C); and cyan triangle, CDBW ($S_A = 34.81$ g/kg, $\Theta = -0.72°$C). Density contours $\gamma^n = 28.18$, 28.22, 28.27, and 27.34 are shown.



As reflected in their relatively small weights, the pseudo-tracers, PO$^*$, NO$^*$, and SO$^*$, do not provide strong constraints as manifested by the almost linear distribution of the data points when they are plotted against temperature (Appendix, Fig.A1).

The number of the unknowns (source water masses) is four; two are CDBW and WSDW, and LCDW which overlies these AABW is expressed by the linear combination of two extremes (warm/salty and cold/fresh), which we call LCDW1 and LCDW2. It is possible that the bottom waters from the Ross Sea (RSBW) and Adélie Land (ALBW) cross the I07S section

after negotiating the Princess Elizabeth Trough, but some previous studies have suggested otherwise. A numerical simulation by Kusahara et al. (2017) indicates that bottom waters that have passed through Princess Elizabeth Trough form a deep eastern boundary current flowing along the west coast of the Kerguelen Plateau, that is, eastward from 60°E (their Figures 8a and 8b) and do not spread into the Enderby Basin. Aoki et al. (2020a) also reported that south of 62°S near the bottom (within 300 dbar of the seabed), RSBW and ALBW influences were discernable only on 70°E and not on 60°E. We therefore neglect

the contributions from ALBW and RSBW to the interior Enderby Basin. We expect that LCDW originates mostly from the upstream ACC. As seen in Figure 9, LCDW in the interior Enderby Basin is not a homogeneous water mass but shows a range of temperature and salinity. We thus chose two end points and express the range as the mixing (linear combination) of these two. One end point is Station 31 (42.5°E, 30°S) depth 4670 dbar ($\gamma^n = 28.265$), and the other is Station 50 (52.0°E, 30°S) depth 3210 dbar ($\gamma^n = 28.279$). For WSDW, we use data from the bottom (5245 dbar) bottle at Station 71 ($\gamma^n = 28.4$) from

the 2008 I06S occupation. CDBW is represented by the C03 station of the KH19 cruise, 61.81°E 63.1°S, sampling depth 4542 dbar ($\gamma^n = 28.394$), because the water properties there ($S_A = 34.81$ g/kg, $\Theta = -0.72°$C) are closest to the average value for CDBW measured by a year-long mooring ($S_A = 34.807$ g/kg, $\Theta = -0.777°$C when converted to the TEOS-10 value) near Cape Darnley (Ohshima et al., 2013).

The mixing ratios in Fig. 10 show some scatter, but the overall composition of AABW varies roughly with density. On $\gamma^n = 235$ 28.27, CDBW=0% to 20%, WSDW=5% to 35%, and LCDW=LCDW1+LCDW2=70% to 95%. The contribution changes approximately linearly towards $\gamma^n = 28.35$ where CDBW=0% to 35%, WSDW=35% to 75%, LCDW=25% to 45%. As expected from the closeness of the I07S temperature and salinity to those of I06S (Fig. 9, left), AABW in the Enderby Basin consists mostly of WSDW with little contribution from CDBW. Near the shelf ($X > 5000$ km, south of 63°S), near-bottom water mass shows considerable contribution from CDBW, as indicated by several isolated relatively fresh points on the temperature-salinity

diagram (Figure 9).

We examined the sensitivity of the results to the choice of source water masses. For WSDW, Station 70 (30.0°E, 62.0°S, 5267 dbar), Station 69 (30.0°E, 61.5°S, 5294 dbar), Station 73 (30.0°E, 63.5°S, 5217 dbar), and Station 75 (30.0°E, 64.5°S, 5105 dbar) were substituted for the original source water at Station 69. Similarly for CDBW, C02 (59.45°E, 63.50°S, 4370 dbar), C04 (63.69°E, 64.10°S, 3922 dbar), C05 (65.20°E, 64.96°S, 3364 dbar), and C06 (66.25°E, 65.50°S, 3363 dbar)

were each substituted. From these combination we obtained 25 pairs of calculations: the spreads of the results at Stations 134 ($L = 4215$ km, 57.22°S), 142 ($L = 4650$ km, 61.09°S) and 148 ($L = 4975$ km, 63.97°S). are shown in Fig. 11 Because LCDW is represented by the linear combination of two extremes, the sensitivity to LCDW was not examined. The estimates for CDBW and WSDW show an uncertainty of $\pm 0.2$ or greater. In contrast, the estimates for LCDW show smaller spread. The uncertainty,




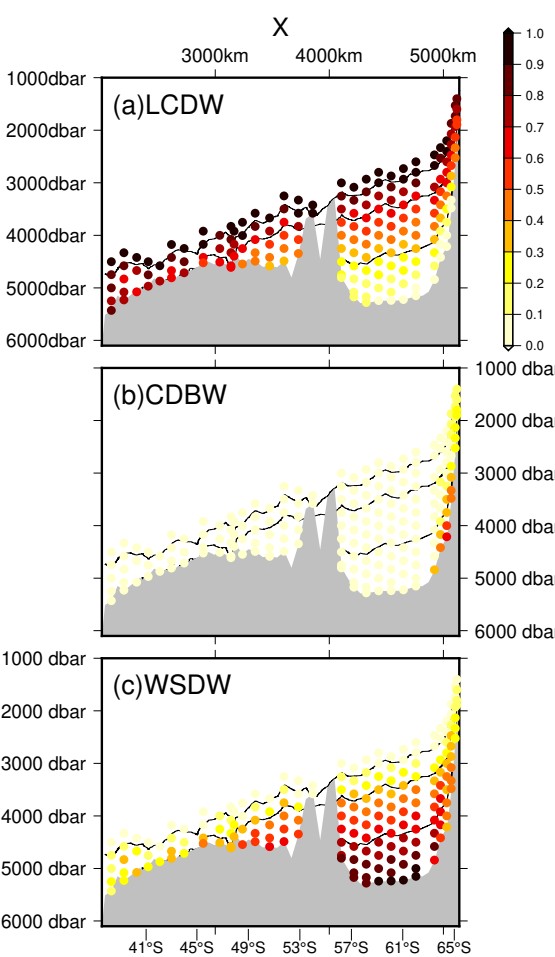

**Figure 10.** Mixing ratio of sources for Antarctic Bottom Water in the Enderby and Crozet Basins along the I07S section. The upper horizontal axis shows distance, $X$, the bottom axis shows approximate latitude. The vertical axis shows depth. The contours are isopycnals at $\gamma^n = 28.27, 28.30$ and $28.35$.



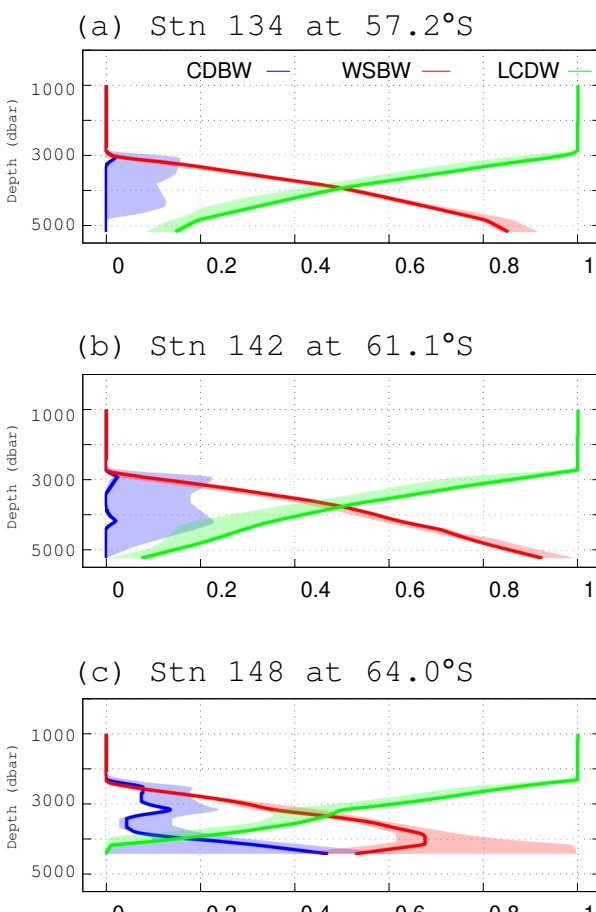

**Figure 11.** Vertical profiles of the estimated composition (horizontal axes) of AABW in the Enderby Basin. The contributions from Cape Darnley Bottom Water (CDBW), Weddell Sea Deep Water (WSDW), and Lower Circumpolar Deep Water (LCDW) are shown by blue, red, and green lines, respectively, and the accompanying shading shows the spread (minimum and maximum) of the estimates for 25 pairs obtained using different choices for CDBW and WSDW properties. The bottle data were vertically interpolated by the method of Reiniger and Ross (1968).





**Table 3.** Summary of water temperature changes $\Delta\Theta$ and salinity changes $\Delta S_A$

| station | latitude | previous | UCDW $\Delta\Theta$ $\gamma^n = 27.7$[a] | LCDW/AABW $\Delta S_A$ $\gamma^n = 28.27$[b] | Comment |
|---|---|---|---|---|---|
| 78 | 32.2°S | Mar 1985 | 0.076 | n/a | |
| 83 | 34.0°S | '87,'02,'09 | n/a | n/a | AAIW and LCDW oscillation |
| 86 | 35.0°S | Mar 1985 | 0.168 | n/a | S-max[c] erosion by 0.01 g/kg |
| 92 | 37.8°S | Mar 1985 | 0.092 | −0.003 | S-max in LCDW increase by 0.02 g/kg |
| 97 | 40.3°S | Mar 1985 | −0.044 | n/a | |
| 103 | 43.0°S | Mar 1985 | −0.175 | −0.002 | S-max in LCDW decrease by 0.01 g/kg |
| 107 | 45.0°S | Apr 1993 | 0.147 | −0.002 | |
| 108 | 45.5°S | Mar 1985 | −0.207 | −0.004 | |
| 113 | 47.7°S | Mar 1994 | 0.303 | −0.006 | Two peaks in 1994 LCDW salinity |
| 115 | 48.0°S | Apr 1991 | 0.110 | −0.005 | |
| 117 | 49.0°S | Apr 1993 | 0.145 | −0.005 | |

a Upper Circumpolar Deep Water (UCDW) is represented at this density. $\Theta$ is Conservative Temperature.

b Lower Circumpolar Deep Water (LCDW) and Antarctic Bottom Water (AABW) are represented at this density.

$S_A$ is Absolute Salinity.

c Local maximum in salinity $S_A$.

however, does not change the qualitative conclusion that the contribution of WSDW to the AABW in the Enderby Basin is
greater than that of CDBW.

## 5   Decadal changes in water mass properties

Eleven stations were reoccupations of previous stations at which high-precision hydrographic data had been otained. The observed changes since the previous occupation are summarised in Table 3, and those at Station 117 and Station 83 are shown in Figs. 12 and 13, respectively. Changes at the other stations are in Appendix (Fig.A2).

UCDW became warmer by $0.09°$ to $0.17°C$ except at the stations between 40.3°S and 45.5°S, where the UCDW water mass is subject to frontal movements and accompanying mesoscale mixing such that the changes are neither systematic nor robust (see also Fig. A1). As pointed out in Section 3, UCDW has a dissolved oxygen front at around 27°S, and UCDW warming observed at stations 86 and 92 might suggest warming of the northern UCDW. Aoki et al. (2005) reported warm and salty anomalies in UCDW, which they attributed to mixing surface waters that were warmer and fresher because of climate changes.
The UCDW warming found here confirms that this warming trend continued after the 1990s.

The changes in the lower water masses, LCDW and AABW, are smaller, but a clear isopycnal freshening signal of $0.005$ g/kg was found at Stations 115 and 117, over the sill connecting the Crozet and Enderby Basins. Within the Crozet Basin (Stations 83 to 108), the isopycnal freshening was smaller, $0.002$ to $0.004$ g/kg. During the 2019 cruise, the accuracy for



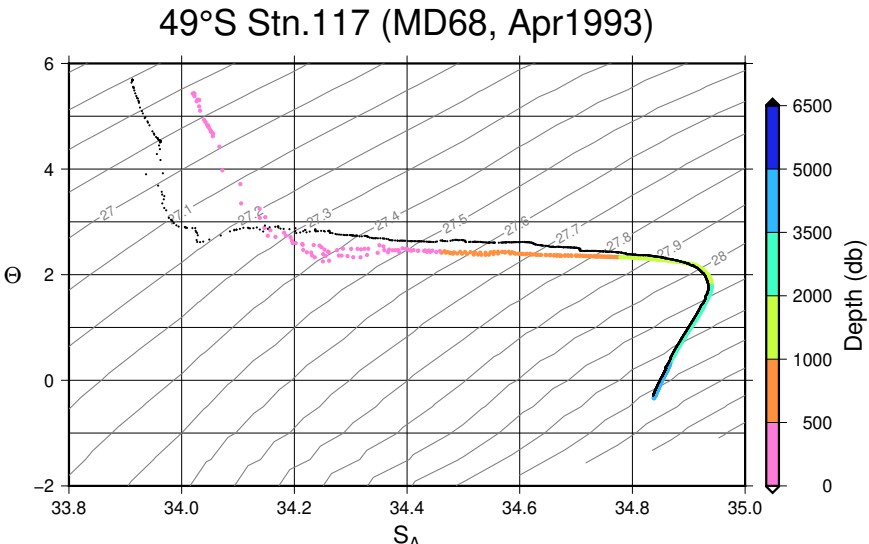

**Figure 12.** Water properties at Station 117 which was occupied during cruise MD68 by M/V *Marion Dufresne* in April 1993 (coloured points; the different colours indicate depth). Black points show the 2019/20 occupation results.

salinity conformed to the GO-SHIP standard of $0.002$ g/kg. The accuracy for salinity during the M/V *Marion Dufresne* cruise
was $0.003$ practical salinity unit (approximately $0.003$ g/kg) (Park et al., 1993). Therfore the observed difference, which is
not significantly larger than the combined uncertainty of $0.004$ g/kg, is not conclusive. We note that the freshening at Stations
115 and 117 (Table 3) was measured on the isopycnal in order to follow the vertical movement of the water masses, but the
interpretation needs some care particularly near the bottom. The deepest observation at Station 117 in 1993 was at 4430 dbar
with $\Theta = -0.337°$C and $S_A = 34.838$ g/kg giving $\gamma^n = 28.340$, whereas the I07S observation at the same station at 4430
dbar was $\Theta = -0.281°$C and $S_A = 34.836$ g/kg at $\gamma^n = 28.328$. Although the observed change on the isopycnal is cooling
and freshening ($\Theta = -0.262°$C and $S_A = 34.842$ g/kg at $\gamma^n = 28.328$ in 1993), it is the well-known volume reduction of
AABW (Purkey and Johnson, 2012) that best explains the observed change. This volume reduction of AABW is consistent
with the findings further west at at 30°E, where no significant salinity change in bottom water away from the continental slope
has been reported (Purkey and Johnson, 2013; Couldrey et al., 2013; Aoki et al., 2020a). Freshening of bottom water off Cape





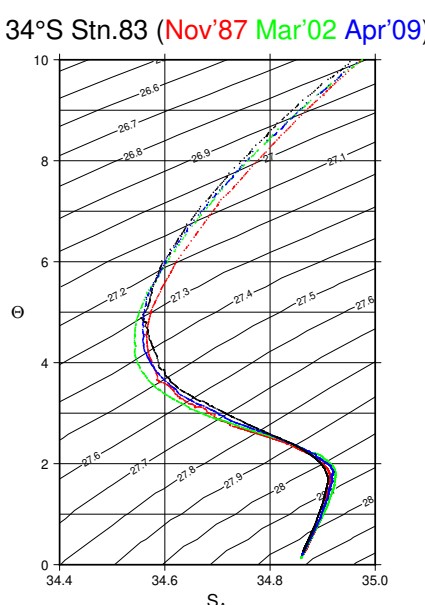

**Figure 13.** Water properties at Station 83, a part of the I05 section previously occupied in November 1987, March 2002, and April 2009. Data from these past occupations are shown by red, green, and blue dots, respectively, and our 2019 occupation are shown by black dots.

Darnley is also weak (0.002 to 0.004 g/kg per decade). As estimated in Section 4.2, CDBW is diluted by a factor of 10 or more around the sill between the Crozet and Enderby Basins, the CDBW freshening likely does not contribute to the observed freshening at Station 115 and 117.

     Bryden et al. (2003) described the water mass changes at the cross point of the GO-SHIP I05 section and the I07S section at 34°S, with a focus in the water mass changes in the thermocline (as defined by potential temperature $\theta_\sigma > 6°C$) or the mode

waters in the northern Crozet Basin, Bryden et al. (2003) reported a freshening of the thermocline water mass (between potential temperatures of 7° and 10°C) from 1987 to 2002. This cooling and freshening was also reported by Aoki et al. (2005). These changes are attributable to surface warming in response to atmospheric warming. The freshening of the thermocline water mass at Station 83 appears have stopped from 2002 until 2009, after which it resumed (Fig 13). This behaviour is different from that around the salinity minimum of AAIW where the freshening from 1987 to 2002 was as large as 0.02 g/kg near the salinity

minimum at $\gamma^n = 27.4$. After 2002 the change reversed and the salinity minimum become saltier through 2009, continuing until 2019. The salinity maximum originating from NADW at $\gamma^n = 28.05$ also shows the oscillation – saltier from 1987 to 2002 then fresher through 2009 until 2019.

     These property changes have often been attributed to meridional migration of the water masses (e.g. Gille, 2002). It is known in this region that the fronts, which are indicators of the meridional position of water masses, move southward on a

decadal timescale (Sokolov and Rintoul, 2009; Kim and Orsi, 2014). Along the I07S section, sea surface height has increased



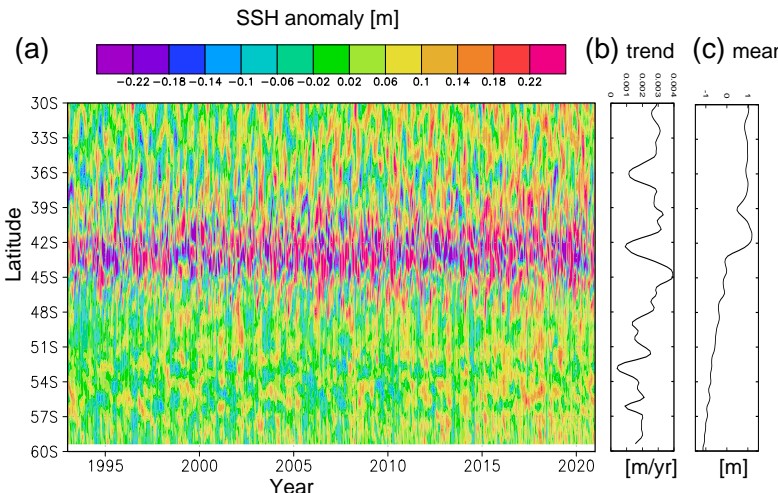

**Figure 14.** (a) Daily time series of the sea surface anomalies measured by satellite altimeter along the $57.625°$S meridian. Least squares fitting at each latitude yielded profiles of (b) trend and (c) mean.

by about $0.003$ m per year north of 48°S (Fig. 14). Given the background gradient of approximately 2 m over 30° latitude, the fronts shifted southward roughly $0.003 \times 30/(2/30) = 1.35°$ of latitude over a period of 30 years. At the typical UCDW depth of $2000$ dbar around 40°S, a close look at Figure 3a shows that $\Theta$ does not show an obvious meridional gradient north of the subtropical/subantarctic fronts at 42°S. South of the fronts, the meridional gradient of $\Theta$ is about $0.1°$C per $1°$ latitude.

A southward shift of the front can therefore explain the warming observed at Stations 113, 115, and 117. Salinity at the LCDW/AABW depth of $4000$ dbar shows a persistent meridional gradient of about $0.0015$ g/kg per $1°$ latitude, but salinity is higher northward. A southward shift of the fronts cannot explain freshening observed at these depths. Thus a southward shift of the fronts across the I07S plausibly contributed to the UCDW warming observed south of 47°S, but other changes cannot be attributed to the meridional water mass movements.

Although the seven southernmost stations south of 64°S have been occupied three times (1994, 2012, and 2019), it is difficult to identify robust trends over the two decades in this near-coastal region, which is subject to direct forcing from both the atmosphere and ice. In this section, we thus have argued only the changes observed in the Crozet Basin.

## 6    Conclusions

In this paper, results from a 2019 cruise along 55°E south of 30°S are delineated with emphasis on deep to bottom circula-
tions. At 55°E, the deep to bottom circulation is primarily diffusive with no obvious mean flow – thus transient tracers show equatorward spread in accordance with the large-scale meridional gradient. In this sense, the eastern boundary of the Weddell Gyre does not reach as far east as 55°E.

    Because the 2019 cruise was the first occupation of the new GO-SHIP section I07S, only 11 previously occupied stations were revisited (Table 3). In the present comparison, the bottom salinity "rebound" in the Australia-Antarctic Basin, which was
reported to be propagating westward (Aoki et al., 2020b) was not detected. The next GO-SHIP occupation of the I07S section will measure changes in the Enderby Basin not only in temperature and salinity but also in other parameters that were sampled during the 2019 cruises, including nutrients calibrated against certified reference materials and biological parameters such as microbial diversity and abundance.

*Data availability.* Hydrographic data from GO-SHIP cruises are available from CCHDO https://cchdo.ucsd.edu. Hydrographic data from
*Marion Dufresne* are available from SeaDataNet https://cdi.seadatanet.org/. Hydrographic data from *R/V Hakuho* are available from https://ads.nipr.ac.jp/dataset/A20240612-002 and https://ads.nipr.ac.jp/dataset/A20240613-001. The altimetry data are from E.U. Copernicus Marine Service Information; https://doi.org/10.48670/moi-00148. In Fig. 2, Scripps Argo float drift data ( https://doi.org/10.6075/J0KD1Z35) were used. In Figure 6 World Ocean Atlas 2023 (Reagan et al., 2024) was used. The atmospheric concentration of CFC12 and $SF_6$ after 2015 was downloaded from NOAA Global Monitoring Laboratory (https://gml.noaa.gov/hats/data.html).

**Appendix: Additional figures**



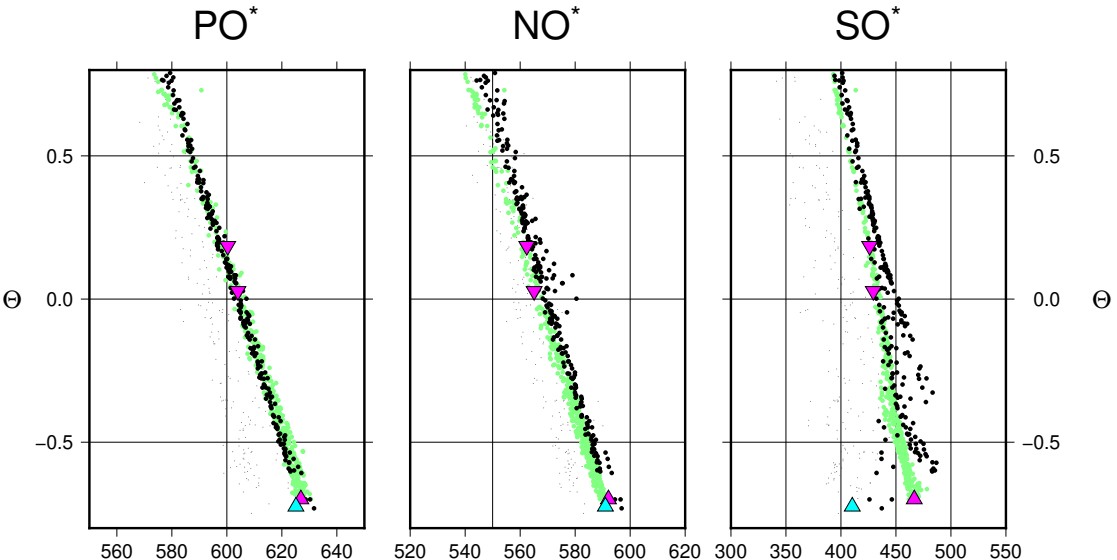

**Figure A1.** Pseudo-tracers $PO^* = 170[PO_4]+[O_2]$ (left), $NO^* = 10.625[NO_3]+[O_2]$ (centre), and $SO^* = [SO_4]+[O_2]$ (right) plotted against conservative temperature. The green points show deep ($\gamma^n > 28.0$) data on I06S (30°E). The black points show deep ($\gamma^n > 28.0$ and depth greater than 3000 dbar) data on I07S. The grey dots show data from the KH19 cruise. The source waters used in the mixing analysis are indicated by triangles; LCDWs by magenta inverted triangles ($PO^* =600$, $NO^*=562$, $SO^*=426$ for LCDW1; $PO^* =601$, $NO^*=565$, $SO^*=430$ for lCDW2) WSDW by the magenta triangle ($PO^* =627$, $NO^*=592$, $SO^*=467$) and CDBW by cyan triangle ($PO^* =625$, $NO^*=591$, $SO^*=410$).





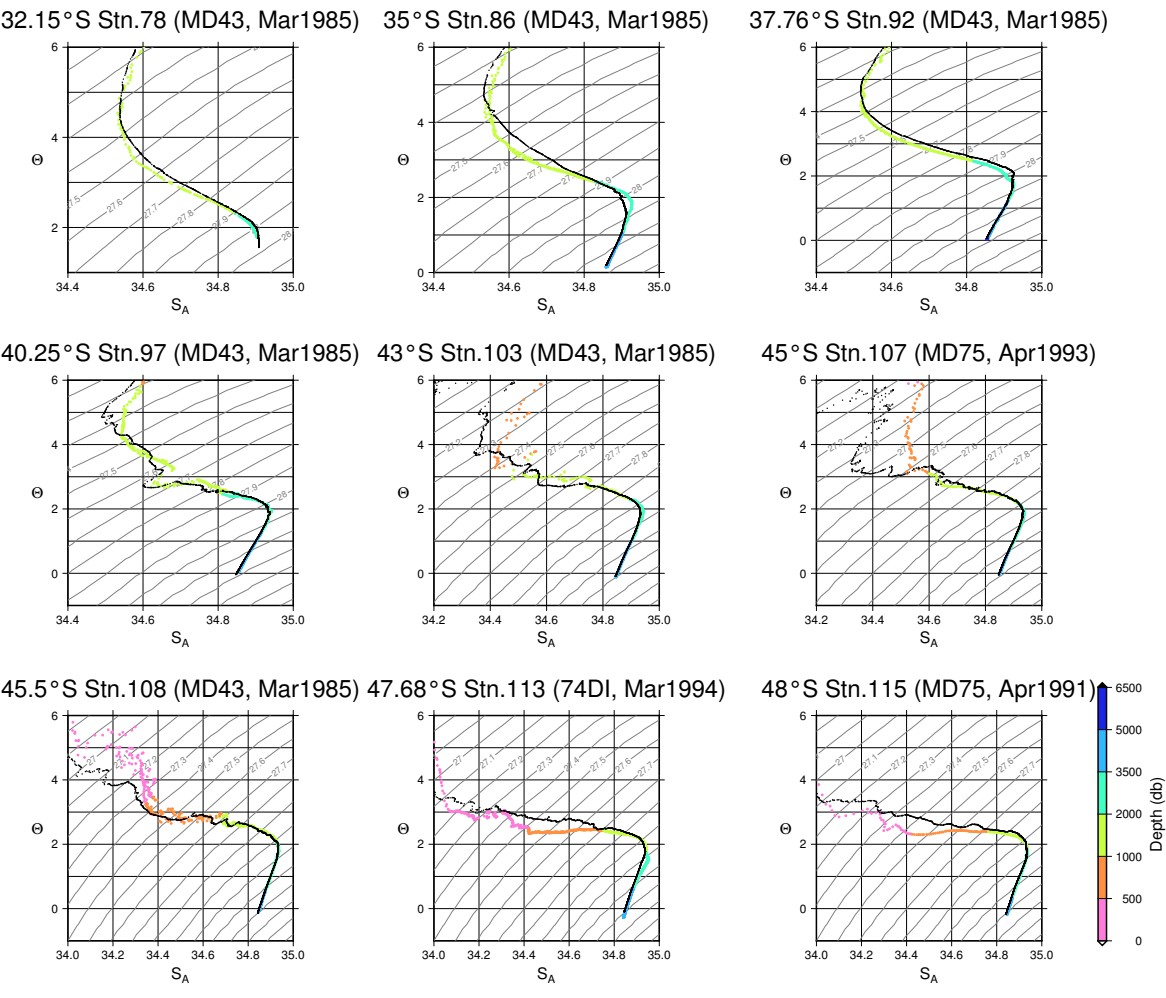

**Figure A2.** Observed water property changes at all stations listed in Table 3 in the main text. Colored points show data from the indicated occupation year and black points show the 2019/20.



*Author contributions.* K.K. participated in collection of the I07S data, wrote software, and wrote the original draft. S.A. conceptualised and provided methodology & interpretation. K.I.O. and M.Y.-K. collected & curated the KH data and contributed in interpretation. All authors contributed in funding acquisition, and reviewed & edited the submitted manuscript.

*Competing interests.* One of the authors is a member of the editorial board of Ocean Science.

*Acknowledgements.* The authors thank the officers and crew of all cruises listed in Table 1.



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
