# Peer review of "Hydrographic section along 55°E in the Indian and Southern oceans"

_EGUsphere, 2024_

## Author Comment (AC1)

Citation: https://doi.org/10.5194/egusphere-2024-2237-RC1

**Reply to Review of the MS "Hydrographic section along 55°E in the Indian and Southern oceans" by Katsumata *et al**

We thank the reviewer for careful reading and constructive comments. In this reply, we will list our main responses, point-by-point. The detailed reply will accompany the revised manuscript. Your comments are in *Italic* and our replies are in Roman fonts.

> *I have three main criticisms, which I believe the authors can solve. One is about salinity changes. Salinity measurements need further corrections when comparing observations from different cruises (see the series of work of Purkey and Johnson), which are particularly critical for the deep/abyssal ocean. I am unsure if the results of salinity changes described in the present MS are robust as they are in opposition with Choi et al. (2022) (see my point below).*

The authors thank the reviewer for bringing up this important matter. Indeed, consideration of the possible offsets in Standard Seawater batches increased the uncertainty significantly and we are no longer certain about the change in salinity that we observed. Those conclusions derived from the salinity changes are now deleted. With consideration for the third point below, the topic of decadal changes seems no longer appropriate in the main text and we have moved the section to Appendix after shortening it.

> *Second, the quality of the LADCP observations has never been mentioned or weighted in some discussions about the deep circulation in the paper.*

This point will be addressed in the revision (see below).

> *The third point is the lack of links between sections. It seems the authors have written different pieces and put them together as MS. A better link between sections would have benefitted the MS. I suggest the authors expand the conclusion/discussion to link the sections and bring some conclusions that move forward the understanding of the deep ocean in the Southwestern Indian Ocean portion of the Southern Ocean.*

In response to this comment, we have added a paragraph in Introduction, which now have subsections to clarify its structure where we explain that the three topics explored in this paper do not have strong inter-connection and the reader may choose to pick up the sections of interest and skip others.

**Line-by-line comments:**

> *In the main text, sometimes it is typed "Figure x," and sometimes "Fig. x." Choose one and use it throughout the text.*

We have found out that "Fig." is the correct form (`https://www.ocean-science.net/submission.html#figurestables`) and changed "Figure"'s accordingly.

> *Table 1 is not cited in the main text. Consider citing it in "section 2. Data" I guess the table lists the cruises analyzed in the present work.*

A sentence is added to refer to Table 1.

> *Table 2: The potential temperature units are missing. Units could be added in the caption or table interior.*

The unit (°C) is added in the caption (rather than in the table in order to compress the horizontal size of the table.).

> *Figure 4: The fronts cited in the text (L#75-80) could be added to the figure to make it easier to identify. Consider using a horizontal axis with latitude, which also would help with interpretation. We never know where stations start counting on a cruise, whether at the southmost or northmost point. Is this from I07S? It would be nice to add to the caption. I suggest changing the vertical black lines for something less overwhelming, such as grey.*

(Fig.4 → new Figure 3) The fronts are added in the upper and lower panels. Latitude ticks are added. The station numbering is now remarked in the Data section. The vertical grid lines are now thinnest grey.

> *L#81: Fig. 3 appears after Figure 4, which is confusing. Consider order figures sequentially as they appear in the text.*

Figs. 3 and 4 are swapped.

> *Figure 3: Consider reducing the amount of vertical black lines that make it harder to identify features (it will be even harder when formatted to the published paper). Also, I didn't get the spacing of tick markers between major ticks at the bottom axis. For clarity, consider removing the minor tick markers. The vertical axis is pressure, not depth, as described in the caption. I couldn't find the triangular shapes mentioned in the caption. Are they plotted? In Section 2, it is not mentioned that I07S and I07N data have been merged (or in the Figure 1 map). Have the salinity and dissolved oxygen of both cruises been cross-calibrated (particularly in the deep ocean)?*

The CTD and XCTD traces are all removed and substituted by tick marks on the upper horizontal axis. The tickmarks and labels for latitudes are now in brown

while the tickmarks for distance remain in black. The caption is rewritten to refer to pressure rather than depth. The triangular shapes appear in our PDF renderer but with the hope that they do not appear in the renderers of the readers, the remark on this triangular shapes were removed. The data for I07N and I07S are just plotted side by side and not used for quantitative analyses, thus not cross-calibrated.

> *L#83: XCTD data is not mentioned in Section 2. Please add.*

A paragraph is added to mention XCTD.

> *L#85: The terminology is not adequate. Both are real fronts, but one is associated with a transient feature (mesoscale eddy) and the other with a permanent feature of the ocean circulation.*

Agreed. Changed to "ACC fronts".

> *L#97-99: The argument is unclear. How do the salty waters transported by the Agulhas Current/Agulhas Return Current amplify P-E meridional gradients? It can enhance the haline gradients but not P-E. Or are you arguing that there is some coupling with the atmosphere and salinity would increase or decrease P or E?*

Changed to "the meridional haline gradient imprinted by precipitation minus evaporation".

> *L#103-104: I suggest breaking it into two statements, as there are two distinct pieces of information that are hard to understand in the current grammar structure. One describes the features encountered in the section, and the other is the vigorous isopycnal mixing.*

Rewritten into multiple sentences as suggested.

> *L#108: If the sections are instantaneous snapshots, how would the LADCP show the mean flow in a region dominated by mesoscale eddies (previously shown)?*

The adjective "mean" does not indicate temporal averaging but designates spatial averaging, i.e., large-scale background flow. Changed "mean" to "large-scale background".

> *L#109: What do you mean by "rich eddies"? Strong eddies? Please re-write for clarity.*

Changed "rich" to "numerous".

> *L#110: I am intrigued by how a snapshot could capture a "mean transport." It is mentioned in the text the LADCP "could not capture any mean transport"... I guess the text is trying to say that there is no coherent large-scale pattern in the LADCP data, and the deep circulation is dominated by mesoscale, which is an interesting*

*result. Please consider re-writing this part. Question: how is the LADCP data quality in the deep ocean? With fewer scatterers in the deep ocean, LADCP-based velocity profiles are sometimes not of good quality. No info about the LADCP data quality in I07S is given in Section 2*

As suggested by "no coherent large-scale pattern" above, we paraphrased "mean flow" to "large-scale background flow". We had two (upward looking and downward looking) sensors and both returned good signals even in the deep ocean. The data quality is now explicitly remarked.

*L#115-189: Section 4.1 Isopycnal diffusivity estimated from transient tracer distributions. How are the diffusivity estimates calculated based on the CFC-12/SF6 tracer distribution related to the estimations based on the fine-scale parametrization calculated in section 4? It is unclear to the readers what the aim of obtaining both estimates is. How does the diffusivity link with the rest of the study?*

The fine-scale parameterisation gives only **diapycnal** diffusivities, while the CFC-12/SF6 tracer gives only **isopycnal** diffusivities. These two quantities are independent in general and one does not necessarily infer the other, thus we believe it is meaningful to give both two estimates.

*Figure 8 b/e and Figure 10: pressure, not depth, as described in the respective captions*

The captions for Figs. 8 and 10 are corrected.

*L#179: It is unclear what the text meant by "is not unlike the deep-sea value." Please re-phrase for clarity. Is this high diffusivity near any specific bottom topography? Or is it associated with high-bottom roughness?*

We meant that the value was typical for deep seas. This isopycnal diffusivity is not particularly high.

*L#190-250: Section 4.2 AABW composition. How does this decomposition relate to the Lagrangian simulations of Solodoch et al. (2022)? Are they consistent?*

Solodoch, A., Stewart, A. L., Hogg, A. M., Morrison, A. K., Kiss, A. E., Thompson, A. F., et al. (2022). How does Antarctic bottom water cross the Southern Ocean? Geophysical Research Letters, 49(7), e2021GL097211. https://doi.org/10.1029/2021GL097211

Solodoch et al. (2022) assumed their artificial tracers at the surface of the shelf regions. On the other hand, our source used in Section 4.2 were sampled near the bottom. Since the dillusion of the surface water at the bottom is not known, it is difficult to compare quantitatively the results from Solodoch et

al. (2022) with out results. Qualitatively, the results do not disagree – for example, the near-bottom water in the Enderby Basin consists of the Weddell and Prydz tracers and negligible contribution from the Ross and Adelie tracers (their Fig.1); vigorous mixing is confined south of 60°S (their Fig.2) and mixing is moderate to weak further north. Our assumption in the mixing analysis that Adelie and Ross waters do not contribute to the Enderby AABW is also consistent with their results.

> *Figures 9, 12, and 13: units for conservative temperature and absolute salinity are missing*

The units are added to Fig 9, 12, and 13.

> *L#252-254: In this section, salinity changes are shown. However, there is no description of salinity corrections applied to the different cruises in Section 2 (e.g., correction for different standard seawater, the ad-hoc correction from Purkey and Johnson, etc.). This is critical when comparing temporal salinity changes in the deep ocean from measurements taken decades apart.*

Agreeing that the reviewer made an important point, we have applied the Standard Sea Water batch correction, where possible. For *Marion Dufresne* cruises, the batch correction was not possible and we had to include the possible offsets into our uncetainty. Given this increased uncertainty, we are no longer confident with the salinity changes and rewritten the discussion and conclusion. The section seems no longer tenable in the main text and thus moved to Appendix.

> *Table 3: Units for conservative temperature and absolute salinity are missing. Since the initial times are distinct for the different stations, it would be much better to express changes by rates (property change per decade), highlighting stations with great changes. The table info confuses the reader as it shows changes in conservative temperature only for the UCDW and salinity only for the AABW/LCDW layer. I only realized that in my third reading. I suggest adding both temperature and salinity for both layers. Another confusing point is the comments. It seems that these annotations are for the authors, not something directly connected with the text. At least, this was my impression. I suggest deleting the column comments or improving the writing there.*

(Table 3 → Table A1) The units are added in the caption. The decadal rates are added for temperature. Since we compare temperature and salinity at the same density, the changes in temperature can be easily calculated from the changes in salinity and vice versa. We have added the explanation. The comment column has been removed as suggested.

> *L#261-265: A possible freshening in the AABW/LCDW layer is discussed here. This freshening is in contrast with salinification,*

*as pointed out by Choi et al. (2022), which also uses hydrographic observations, but the I06S and a few other cruises. However, the present MS does not mention Choi et al. (2022). Why are the results so contrasting? Would it be due to differences in methodology to calculate changes? Would there be a (likely) lack of corrections for salinity measurements in the present work? Would it be ocean dynamics? The fact is that the changes led to distinct conclusions in the two papers.*

Choi, Y., & Nam, S. H. (2022). East-west contrasting changes in southern Indian Ocean Antarctic Bottom Water salinity over three decades. Scientific Reports, 12(1), 12175. https://doi.org/10.1038/s41598-022-16331-y

As mentioned above, we have removed out discussion regarding the salinity changes as a result of the introduction of the SSW batch correction.

---

## Author Comment (AC2)

Citation: `https://doi.org/10.5194/egusphere-2024-2237-RC2`

**Reply to Comments from Anonymous Referee # 2**

We thank the reviewer for careful reading and constructive comments. In this reply, we will list our main responses, point-by-point. The detailed reply will accompany the revised manuscript. Your comments are in *Italic* and our replies are in Roman fonts.

> *The results are a bit spread between different features: surface fronts, deep layers of the Enderby abyssal plain with a diffusive model; changes in time of deep and bottom water masses in this sector of Antarctica, which is more rarely visited than others or very close to Antarctica.*

This point has also been raised by Referee # 1. In response, one of the topics ("decadal changes in water mass properties") is moved to Appendix. We also added subsections to Introduction and added subsection "Organisation of the paper" where we explain that the three topics explored in this paper do not have strong inter-connection and the reader may choose to pick up the sections of interest and skip others. As a result of this notes, we feel it better to briefly mention the summary of findings in each topic.

> *There is actually a repeated station to the northeast of the sector at 56.5S/63°E (OISO station 11). I am not exactly sure of its bottom depth, but over 4900 db, which bottle data are regularly placed in the GLODAP archive.*
>
> *The paper's reference is*
>
> Ocean Sci., 16, 1559–1576, 2020, `https://doi.org/10.5194/os-16-1559-2020` "Variability and stability of anthropogenic CO2 in Antarctic Bottom Water observed in the Indian sector of the Southern Ocean, 1978–2018", Léo Mahieu, Claire Lo Monaco, Nicolas Metzl, Jonathan Fin, and Claude Mignon.
>
> *This may not interest as much the authors, as it is mostly T, S, O2, DIC, TA, and sometimes, NO3, Silicates (rarely PO4). On the other hand, it could be nice to check some of the trends mentioned in the last section.*

When we search the GLODAP database, we had a criteria that the station location be within 20 km from our stations such that we do not worry about the variability caused by the difference in space and focus in the variability caused by the temporal difference. The OISO station 11 did not meet the criterion. This criterion is now added in Appendix A.

*In the same biogeochemical community, there is an other paper by:*

*Zhang, S., Wu, Y., Cai, W.-J., Cai, W., Feely, R. A., Wang, Z., et al. (2023). Transport of anthropogenic carbon from the Antarctic shelf to deep Southern Ocean triggers acidification. Global Biogeochemical Cycles, 37, e2023GB007921.* `https://doi.org/10.1029/2023GB007921`

We have looked up the reference but could not find a straight-forward link to our manuscript.

*I am a little bit wondering of the interest of the frontal description and the two plots on figure 5 (I believe that one is enough), but as it is not the core of the paper, I dont mind that it is discussed in that part..*

The reason why we have two plots on Fig.5 is to show the daily movements of the eddies.

*When considering the diffusive model, as well as for the water mass composition of the bottom water, the set of constrains is not that large (as clearly some of the variables used are very cross-correlated, as discussed in the appendix). Thus, the choice is made not to take into account the two AABW water masses that originate from further east, as the authors argue that this water does not make it as far west as this section, as it seems mostly flow as an eastern bonderay current northwards in the eastern Enderby Basin. This prompts my comment: If the diffusive hypothesis is relevant, shouldn't it also include diffusion from the eastern boundary (with the water probably having slight different properties). How are you sure that this does not happen? I am just concerned of the limits of the diffusive interpretation and water mass origin made in the paper (not so sure that it would change much, in fine!). Of course, I am aware that you only have a meridional section, and thus not the zonal variability component within the Enderby basin (the other two sections are further away which makes sense!)*

We admit that the diffusive effects from the eastern boundary current along the Kerguelen Plateau is beyond the capability of our simplistic model. A remark is added.

**Minor comments:**

*My other comments are mostly on some details that could be improved or on which I had minor questions.*

*Fronts on map 1 a bit strange east of Kerguelen-McDonald plateau and Fawn Trough (mostly SACCF not getting in right place?), but*

*no importance for the topic and front names could be overlaid on contours for example in the west of the map (where they are all well separated). Hard to see little crosses, circles, that are small (and at 30°E overlaid on longitude line)*

Our definition of SACCF follows that of Orsi et al. (1995, see their Fig.7). The front names are now printed along the contours except for SAF and SACCF (found no spece for these two). The size of the marks (crosses, circles) is a compromise; if too large, the stations overlap. The 30°E longitude line is removed.

*42, evidence for eddy activity at 1000 dbar (reference on the product not reported in the text, but in the figure 2 caption, where it seems to 1°x1° mapped Scripps Argo drift data (Katsumata, 2017); would the ANDRO (French) product show the same features?). At first hand, I was surprised that, on figure 2, EKE seems larger at 1000m (but that might be some filtering in the altimetry data). Contours on figure 2 hard to visualize (the 200m and 5000 m contours should be done with different colours)*

We did not try plotting the same figure from ANDRO (`ftp://ftp.ifremer.fr/ifremer/argo/etc/coriolis-custom/argo-andro-data/`), but ther document (Ollitrault and Rannou, 2013, `https://doi.org/10.1175/JTECH-D-12-00073.1`) convinces us that the plot will be similar.

As the title of the figure shows, the EKE at 1000 m is multiplied by 10. We added this to the caption. The contours are now in different colours (black and brown).

*76 Figure 4 seems to be cited before figure 3 (l. 81). I have also some difficulties seeing the colour curves (too thin) on the lower panel of figure 4 (also, the colour on top panel)/*

The order of Figs. 3 and 4 are changed. The color on (new) Fig.3 are changed for better visibility.

*78: E instead of S for two latitudes.*

Corrected.

*Figue 3: in the sections, it seems that there is no station to the bottom near 4000 km. This could be be mentioned in describing the data (as this is one area, where the horizontal resolution indicated is not reached, except in the top 2000 db (XCTDs?))*

Description of XCTDs were missing in the original version. It is now added.

*Figure 6: gradient reported at $\frac{1}{4}$ degree grid, but results from some spatial smoothing in WOD2023 (typically, on the order of 3°). The maximum gradient reported on line 96 at this location might be due to the stationarity of the front at this longitude. I don't think that the 'instantaneous' gradient is weaker, for example, at locations further east in the Indian Ocean.*

Agreed. A sentence is added to point this out.

*l.97: 'sality' should be 'salty'*

Corrected.

*105: 'mesoscale structures at 3000 dbar depth'. I am not sure what is exactly refered to. It is not that clear on Q and S sections (at least to the naked eye). What there is is in O2 some strong spatial variability in this region (and depth). Is there some indication from the current measurements of mesoscale structures at this depth and location. I am not so sure that this is indicative of vigorous isopycnal mixing. Or, at least, what the reasoning for that should be explained.*

The 'mesoscale structures at 3000 dbar depth' were not found in temperature nor in salinity – probably because background gradients are weak in temperature and salinity compared to that of dissolved oxygen. The horizontal currents measured by LADCP showed a turbulent flow similar to those shown in Figure 7. We have added this comment.

*Figure 7 shows currents integrated over neutral density range 27.9 to 28.27. Unfortunately, figure 3 does not show 27.9 (it starts contours at 28.0). Where is this neutral surface located (or could we instead show currents in 28.0-28.27 ofr LCDW layer?). On this figure does one have an idea of the uncertainty in the velocity profile reconstruction. In particular I was a little puzzled by the strong northward velocity component in LCDW for the northern stations in the basin part of the section. Surprising in the two ellipses presented, it seems that the residual average current is exactly zonal. I find that really surprising, and wondered whether the meridional component is not plotted. I am also not sure on how to read the scale of the vectors presented on the figure. I would help to have an arrow below the plot with its velocity value to report this information.*

The top density contour on (new) Fig.4a is changed from 28.0 to 27.9. The accuracy of LADCP horizontal velocity is difficult to estimate (see, e.g. Polzin, K. L., E. Kunze, J. Hummon, and E. Firing (2002), The finescale response of lowered ADCP velocity profilers, J. Atmos. Oceanic Technol., 19, 205–224). We added, at least, a description of good LADCP data quality. Hints for "the strong northward velocity component in LCDW for the northern stations in the basin part of section" can be found in right panel of Fig.5, where an cyclonic (therefore clockwise in the southern hemisphere) eddy is found around (58°E, 44°S). The strong northward velocity follows this height contour, thus geostrophic. The averaged flow is not exactly zonal (meridional transport of 0.78 $m^2s^{-1}$ compared to the zontal transport of 14.24 $m^2s^{-1}$ for LACDW. The values are $-27.72$ $m^2s^{-1}$ and $-1.07 m^2s^{-1}$, respectively, for AABW). The meridional component of the transport almost evenly between northward (positive) and southward (negative) transports such that the average is very small. The quantity plotted

on the figure is transport (velocity times thickness thus in $m^2 s^{-1}$), not velocity ($ms^{-1}$). The transport for the vector in the ellipse is found below the ellipse. We made the label larger.

> *113: is the range $10^{-5}$ to $10^{-4}$ $m^2 s^{-1}$ the overall range for all stations, and all depths, or has there been some smoothing. It would be interesting to see its average profile (with quantiles (maybe 20 and 80%) added to see how significant is the near bottom enhancement.)*

The overall range is wider – it is $10^{-6}$ to $10^{-3.5}$ $m^2 s^{-1}$. The diffusivity has been estimated by the internal wave parameterisation such that some smoothing is inevitable (roughly the size of the vertical binning of the spectrum estimation, i.e. 320m in depth). The average profile is found in Figure 4 and the bottom enhancement is clear in Figure 5, respectively, of the original reference, Sasaki et al. (2024) `https://doi.org/10.1029/2023JC019847`.

> *l.115: Isopycnal diffusivity estimated from vertical diffusivity? (and/or tracer distribution).*

The equation (1) shows that the isopycnal diffusivity ($K$) and vertical diffusivity ($D$) are needed. The tracer distribution ($c$) can determine only one of them. We know $D$ from a method independent of the tracer distribution so that $c$ can be used to estimate $K$.

> *l.120-125: here D dependency with depth commented earlier is neglected. This could have some impact on the distribution of tracers and their evolution (as well as one the interior vertical velocity), but maybe it would be a small effect. Can the authors quantify it? Later, I got puzzled as on line 166, mention is made of the spatial variability in diffusivity attributed to figure 8c (but not found on it?)*

It was not possible to estimate the effect of depth dependency of $D$ from this method, as it is treated as constant. If we have more data (say, distribution of independent tracers such as Helium isotope), we might be able to include the effect of the spatial variability of $D$) but for this simplistic model discussed here, $D$ is assumed constant. The variability of $D$ is thus additional uncertainty not included in our estimate of $K$. We have added a remark.

> *Figure 8: different convention on distance than in earlier figures (with 0 at southern boundaries). This is no problem for me, but maybe some readers might be a bit surprised.*

This is why we used a different label ($Y$ here, while $X$ was used in Fig.4 $\rightarrow$ Fig.3).

> *Then description of the mechanistic diffusive model. I am a little skeptical, as with the two sources they prescribe, it seems to me that there are too many parameters, and many approximations (such as dilution over shelf when waters formed, with 50% sounding a bit high, even with that specified there are 5 unknowns to specify). On the other hand, diffusivity values are reasonable, so are the a values.*

We admit that the model might be an over-simplification of the real ocean, but we also found it interesting that such a simple model could produce a reasonable value and decided to report it.

> *After, watermass study for gamma > 28.27. Among equations, they have PO\*, NO\* and even SO\*, in addition to T and S (I am wondering how independent are the different constraints; actually this is presented in Figure A1, and indeed they are highly inter-related and linear with T). Analysis based on Johnson (2008).*

> *On figure 9 caption, mention of 107S, but it does not seem that the plot was retained.*

The data from I07S is found as black dots on both plots.

> *Fig. 9: the left panel is hard to follow with the overlaid data from R/V Hakuho cruises and from the 107S 2019/2020 stations. The two LCDW waters specified on plot are mentioned on line 218. What sets the choice of these two values?*

They are the two extreme (warm & salty vs cold & fresh) stations from I08S cruise in 2008.

> *224: '... only at 70°E and not at 60°E'.*

Corrected as suggested.

> *In table 3 caption, mention changes relative to what.... As is it is not clear what is presented (appears in the text of section 5, but should also appear in the caption).*

The caption to Table 3 is modified to add this information (now Table A1).

> *am also not sure why the change in temperature in LCDW/AABW is not presented on the table (I realize that some of the reported changes (stations 115 and 117) are taken on an isopycnal (which should also be mentioned in the table; this is somewhat different than for the other stations)).*

The caption to Table 3 (new Table A1) is now includes "on isopycnals" to indicate this. See also added new paragraph well as Eq.(A1).

> *291, Figure 14 also suggests some SSH increase further south. I guess that the comment on southward motion refers to the dipole in trend between 42°S and 45°S. However, overall, I appreciate the discussion of trends.* This discussion is now found in the Appendix.

---

## Author Response (AR2)

**Reply to Comments from Anonymous Referee # 2**

The comments are in *Italic* and the replys are in Roman fonts. The line number refers to those in the revised manuscript, not in the track change.

> *I thank the authors to have done a very good job in improving the clarity of the figures as well as adding more explanatory figure captions, and usually adequately modifying the draft. I am however not fully satisfied by two of the replies provided.*
>
> *The first on relating to OISO station 11: 'This may not interest as much the authors, as it is mostly T, S, O2, DIC, TA, and sometimes, NO3, Silicates (rarely PO4). On the other hand, it could be nice to check some of the trends mentioned in the last section.' This answer ('not included as at more than 20 km') is not very satisfactory. I don't think that the data need to be included, but cross-referenced / compared, in case this is relevant (trends can be estimated from those data, and have been estimated). If it is not relevant (too different water masses, for example), this would need to be provided in the response.*

We compared only temperature and salinity (see Appendix A), not oxygen, nutrients and carbons. For temperature, Purkey and Johnson (2010, Journal of Climiate, https://doi.org/10.1175/2010JCLI3682.1) estimated that the horizontal decorrelation length is approximately 160 km. OISO station 11 (56.5°S, 63°E) is much farther aprt from our section than 160 km and it seems impractical to compare temperature between these two stations. The same reasoning applies to salinity.

> *Also, on the comment to the 'Andro' Argo current product: 'We did not try plotting the same figure from ANDRO, but ther document (Ollitrault and Rannou, 2013) convinces us that the plot will be similar'*
>
> *This is reassuring. I have recently been alerted that the number of data included in the SIO product and the Andro product are rather different (more in Andro, at least in the updated version). This points probably to differences in the data selection/processing/quality control. Thus, it might be good to mention that this has been checked with similar results.*

For reference, we now made the plot. We do not know the reason but generally Andro shows weaker EKE's. Since the distributions are similar and does not change our statements about the spatial contrast of EKE at 1000 dbar (LL.39-41), we think it is beyond our scope to discuss the difference in this manuscript.

[Figure]

Figure 1: Eddy kinetic energy at 1000 dbar with two products of Argo drift velocity.

> *Minor editorial comments on revised manuscript: I noticed two typos that need to be corrected in the revised manuscript with track changes:*
>
> - *l. 76 '... station numbers...'*
> - *l. 79: '... to attempt...'*

We thank the reviewer for careful reading.

**Reply to Comments from the Editor**

The comments are in *Italic* and the replys are in Roman fonts. The line number refers to those in the revised manuscript, not in the track change.

> *At many places in the text: Positions do not get parentheses.*

The parentheses were removed.

> *I think the subsections in the introduction are not common and in this case also not necessary.*

Subsections in the introduction were removed.

> *L19-20 "The GO-SHIP (Global Ocean Ship-based Hydrographic Investigations Program) section I07S is the latest occupation." Reading this I expected to read the year of occupation.*

Changed to "The GO-SHIP (Global Ocean Ship-based Hydrographic Investigations Program) section I07S is one of these sections and occupied for the first time in the austral summer of 2019/20." (L.18–19)

> *L41 Not sure if "negotiate" is the correct word here. L238 again "negotiating" seems not the right word here. It is not clear what is meant*

Before submission, the manuscript had been checked by native English editing services. None of them have left a comment here. I have also looked up a dictionary (Oxford Advanced Learner's Dictionary) and "negotiate (v) 3.(formal) to successfully get over or past a difficult part of a path or route" applies here (L.39). We chose to change the latter to "flow through", however (L.238).

> *Section 1.2 about eddies does not seem to belong in the introduction. It shows results of your work, which is not very usual. This might find a better place in the results section.*

This figure is a reproduction of a past work done elsewhere (Katsumata, 2017) and we would rather not include this result as part of the present research which targets the outcome of the cruise. This figure is referenced from the 3rd paragraph of Introduction (LL.39–43).

> *Section 2 Data: Additional data are shown in this paper, namely DO and transient tracers. Please add information about these in this section, like methods, precision/accuracy.*

We have also used nutrients. Relevant remarks are added (LL.70–74).

> *Caption Figure 4: conservative temperature, absolute salinity and dissolved oxygen do not get capitals*

Conservative Temperature and Absolute Salnity need to be capitalised. See Editorial at `https://doi.org/10.1175/JPO-D-13-082.1` in *J.Phys.Oceanogr.* volume 43 issue 5. We agree with the editor that DO should not be capitalised and corrected accordingly (Caption to Fig.4).

> *L280-285 Please check whether the different data providers would request a fair data use statement*

We do not exactly understand what the editor means by "fair data use statement" but we have double-checked "how to cite"s for each database and followed the instruction therein (LL.280–287).

- *L144 flank instead of flak (typo)*
- *L170 delete: unit*
- *L192 delete one: are*
- *L385 Reference Johnson 2008 is incomplete*
- *L402 Please add volume and ages: vol 54, pages 1105-1120*
- *L449 Please add volume and paper number: 129, e2023JC019847*
- *L413-414 delete: "BROKE-West" a Biological/Oceanographic Survey Off the Coast of East Antarctica (30-80°E) Carried Out in January-March 2006, 2010"*

We appreciate the editor's careful reading. All these typos are corrected.

This is such an embarrassment but the first line of the abstract "December 2018 to January 2019" should have been "December 2019 to January 2020". We also found a coefficient "1.66" was missing in front of $[SO_4]$. These were corrected.